# Confidence controls perceptual evidence accumulation

Tarryn Balsdon [1,2✉], Valentin Wyart [1,3] & Pascal Mamassian [2,3]

Perceptual decisions are accompanied by feelings of confidence that reflect the likelihood that the decision was correct. Here we aim to clarify the relationship between perception and confidence by studying the same perceptual task across three different confidence contexts. Human observers were asked to categorize the source of sequentially presented visual stimuli. Each additional stimulus provided evidence for making more accurate perceptual decisions, and better confidence judgements. We show that observers' ability to set appropriate evidence accumulation bounds for perceptual decisions is strongly predictive of their ability to make accurate confidence judgements. When observers were not permitted to control their exposure to evidence, they imposed covert bounds on their perceptual decisions but not on their confidence decisions. This partial dissociation between decision processes is reflected in behaviour and pupil dilation. Together, these findings suggest a confidence-regulated accumulation-to-bound process that controls perceptual decision-making even in the absence of explicit speed-accuracy trade-offs.

[1] Laboratoire de Neurosciences Cognitives et Computationnelles (Inserm U960), Département d'Études Cognitives, École Normale Supérieure, PSL University, 75005 Paris, France. [2] Laboratoire des Systèmes Perceptifs (CNRS UMR 8248), Département d'Études Cognitives, École Normale Supérieure, PSL University, 75005 Paris, France. [3] These authors contributed equally: Valentin Wyart, Pascal Mamassian. ✉email: tarryn.balsdon@ens.fr

 1

Sensory perception results from inferring the causes of uncertain sensory evidence[1]. The perceived objects are under-constrained by sensory evidence, so these inferences are fundamentally probabilistic. In recent years, there has been a growing interest in the ability of human observers to estimate the validity of their perceptual decisions. This form of metacognitive judgement can be obtained by asking an observer to rate how confident she is that one of her perceptual decisions is correct. While the perceptual decision is used to quantify the percept itself, each confidence rating quantifies the certainty the observer has about her own percept. These two types of judgements are known as Type-I and Type-II decisions, respectively[2].

By definition, an ideal Type-II decision-maker uses the exact same evidence as for the Type-I decision[3]. The evidence for the Type-I decision can be computationally described by sequential sampling processes, or diffusion models[4–7] (for a recent review[8]), wherein, samples of noisy evidence are accumulated over time and a decision is made once the evidence reaches a bound. The Type-II decision estimates the likelihood that the Type-I decision is correct, given the accumulated evidence. The likelihood of the Type-I decision is moderated by both the quantity of evidence (determined by the relative placement of the decision bound) and by the quality of evidence (which is marred by suboptimal accumulation, such as noise and leak in the accumulation process). The ideal Type-II decision therefore requires the estimation of the quantity and quality of Type-I evidence.

However, a large body of evidence demonstrates that human observers do not make their Type-II decisions in accordance with this ideal Type-II decision maker. For example, observers poorly incorporate estimates of sensory noise into their Type-II decisions (resulting in over- or under-confidence[9–11]), and they may ignore evidence in favour of other decision alternatives[12–14] (although not always[15]). Furthermore, observers may integrate additional evidence into their Type-II decision that was not used for making their Type-I decision[8,16], allowing them to report errors in the absence of feedback[17–19], and to change their mind after the initiation of a response[20]. These departures from ideal are yet to be fully characterised by a unifying framework of Type-II decision-making.

The computational description of Type-II decision-making is significantly constrained by the tight concomitance of Type-I and Type-II decisions. In the context of sequential sampling processes, the suboptimalities affecting Type-II decisions could be merely inherited from the suboptimalities in Type-I evidence accumulation, or there could be additional suboptimalities in the computation of uncertainty over the Type-I variables. Moreover, there is the possibility that the processes of accumulating Type-I and Type-II evidence are neither identical nor functionally independent, that is, confidence may interact with the very process of accumulating evidence for perceptual decisions. Exploring this possibility is essential for the understanding of confidence and perceptual decision-making.

To clarify this relationship, we asked observers to make the same Type-I judgement but in three distinct Type-II contexts. In all three contexts, the Type-I judgement required observers to make a two-alternative categorisation decision after viewing a series of visual stimuli (a variation of the weather prediction task[21,22]). Based on the work of Drugowitsch et al.[22] each orientation offered a specific amount of evidence in favour of each category, such that the quality and quantity of Type-I evidence can be carefully monitored over the course of each trial. This allowed for the disambiguation of different suboptimalities affecting observers' Type-I decision-making using computational modelling. Given the suggestion that Type-II evidence accumulation may continue after the Type-I decision bound has been reached, we were especially motivated to understand the

relationship between Type-I and Type-II decision-making relative to the point at which the Type-I evidence crosses the decision bound. In the first Type-II context we therefore measured observers' ability to set and maintain appropriate bounds on their Type-I evidence accumulation by asking them to make their Type-I judgement when they thought they had reached an instructed target performance level. We used the second Type-II context to measure observers' default bound—i.e., how much evidence each observer feels they need to accumulate to commit to a perceptual decision. Last, in the third Type-II context, we tested whether observers implement a covert bound when they are presented with more evidence than needed to reach their default bound, as measured in the second context. In this third Type-II context, observers also rated their confidence that their Type-I decision was correct on each trial—i.e., an explicit Type-II decision. In addition to behavioural responses, pupil dilation was monitored throughout the experiment, as the literature has suggested strong links between pupil dilation and Type-I[23], and Type-II[24,25] decision-making via pupil-linked dynamics of the noradrenergic system[26,27]. Both behaviour and pupillometry reflect a partial dissociation between Type-I and Type-II evidence, which could allow for a confidence-regulated accumulation-to-bound process that controls perceptual decision-making even in the absence of an explicit speed-accuracy trade-off.

## Results

**Preliminary analyses.** Across all three Type-II contexts, observers ($N = 20$) made the same Type-I decision: whether the orientations of the sequence of Gabor patches presented on each trial were drawn from the orange or the blue category. These categories were defined by circular Gaussian probability distributions over the orientations of the Gabor patches, as shown in Fig. 1a (see Methods). The three Type-II contexts were presented across two sessions and are depicted in Fig. 1b. For each observer, 100 trials of up to 40 stimuli were pre-defined and repeated six times in the Stopping task context and three times in the Free task and Replay task contexts.

In the first session, observers were placed in a Stopping task context in which they were continually shown samples until they entered their Type-I response. Importantly, they were asked to enter their response when they felt they had accumulated enough evidence to reach a certain probability of being correct (target performance). There were three target performance conditions (70%, 85% and 90% correct) in which observers scored an average proportion correct [95% between-subjects CI] of 0.72 [±0.018]; 0.80 [±0.021]; and 0.82 [±0.023]. This corresponded to a Type-I sensitivity ($d'$) of 1.2 [±0.17], 1.7 [±0.14], and 1.9 [±0.19] in each target performance condition, which was found to significantly increase across target performance conditions using a Wilcoxon sign rank test; $Z$ (70% vs. 85%) = 3.78, $p_{bonf*2} < 0.001$; $Z$ (85% vs. 90%) = 2.35, $p_{bonf*2} = 0.037$, with these $p$-values Bonferroni corrected for two comparisons). Observers also chose to enter their response later in the higher target performance conditions (average median sequence length = 6.5 [±1.13]; 11.5 [±2.09]; and 15.0 [±3.02]; $Z$ (70% vs. 85%) = 3.71, $p_{bonf*2} < 0.001$; $Z$ (85% vs. 90%) = 2.80, $p_{bonf*2} = 0.010$; additional analyses are provided in Supplementary Note 1 and Supplementary Fig. 1).

In a second session, completed on a separate day, observers were placed in a Free task context, followed by a Replay task context. In the Free task context, observers were also continually presented with samples until they entered their response. Unlike in the Stopping task, observers were not given specific performance targets, but instead asked to enter their response as soon as they 'felt ready'. Observers scored an average proportion correct [95% between-subjects CI] of 0.80 [±0.025],

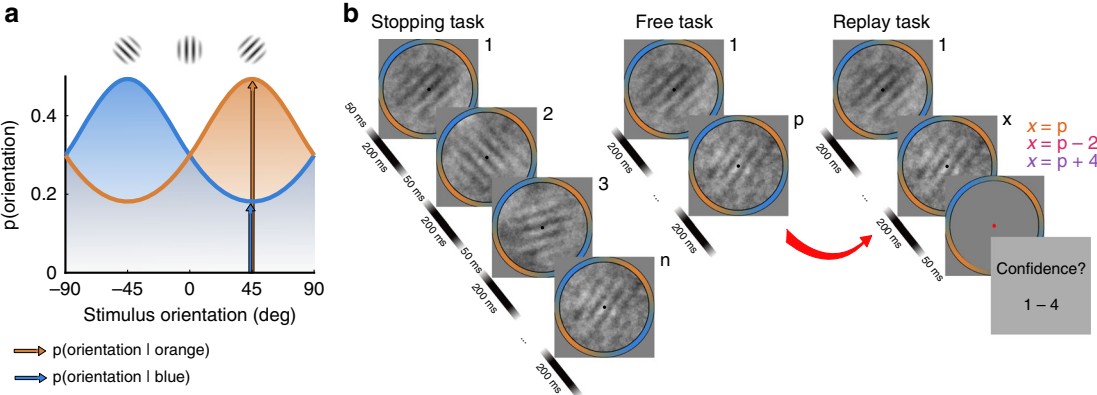

**Fig. 1 Procedure.** On each trial, observers were shown a series of oriented Gabors and had to determine which distribution the orientations were drawn from. **a** The orientations were drawn from one of two circular Gaussian (von-Mises) distributions centred on −45° (blue) and + 45° (orange) relative to vertical (0°). The distributions were overlapping such that a stimulus oriented 45° from vertical is most likely to have been drawn from the orange distribution (orange arrow) but still could have been drawn from the blue distribution (blue arrow). **b** The experiment involved two sessions. In session 1, observers completed the Stopping task. On each trial of the Stopping task the series of stimuli continued until observers entered a Type-I response: blue or orange distribution. They were asked to enter their response when they felt they had a certain probability of being correct. The three target performance levels (70%, 85% and 90% correct) were completed in separate blocks. In session 2, observers completed the Free task followed by the Replay task. The Free task was the same as the Stopping task, except observers were asked to enter their response when they felt ready (after *p* samples). In the Replay task, observers entered their Type-I response once cued (fixation changing to red). They then gave a Type-II response: a rating (1 to 4) of how confident they were that they were correct. Unbeknownst to observers, the Replay task actually replayed the exact same trials from the Free task, except the number of samples was either the same as they had chosen to respond to (*x* = *p*), two fewer (*x* = *p* − 2), or four additional samples (*x* = *p* + 4). These trials comprise the Same, Less and More conditions, respectively. Across all tasks, the fixation point (black dot) and colour guide were present throughout each trial. The samples were presented at a rate of 4 Hz, with 200 ms of stimulus presence (including 25 ms ramp at onset and offset) and 50 ms inter-stimulus interval. All tasks consisted of repetitions of the same 100 trials pre-defined for each observer.

with observers choosing to respond after 10.5 [±1.51] samples on average. The same trials were then repeated to observers in the Replay task, completed immediately after the Free task, but in the Replay task observers were presented with a fixed number of samples and could only respond after the response cue. The number of samples presented on each trial was determined relative to how many samples the observer chose to respond to for the three repetitions of each pre-defined trial in the Free task. There were three intermixed conditions: Less (−2 samples from the minimum), Same (same number of samples as the median), and More (+4 samples from the maximum). This resulted in a median number of samples of 5.4, 10.4, and 18.5 in the Less, Same, and More conditions, respectively. We compared performance in the 100 trials of the Less, Same, and More conditions to the corresponding sets of 100 trials in the Free task exhibiting the minimum, median, and maximum number of observed samples across the three pre-defined trial repetitions. Performance in the Same condition was on par with performance on those same trials (the trials with the median number of samples) in the Free task (mean proportion correct = 0.80 [±0.022]; *Z* (Same vs. Free *d′*) = 0.82, *p* = 0.41, uncorrected). In the Less condition, two fewer samples corresponded to a substantial decrease in performance within-subjects (Less *d′* = 1.01; Free *d′* on Less trials = 1.50; Mean within-subjects difference = 0.49 [±0.13], *Z* = 3.51, $p_{bonf*3}$ < 0.001) but in the More condition, four additional samples did not significantly improve performance within-subjects (More *d′* = 1.85 Free *d′* on More trials = 1.77; Mean within-subjects difference = −0.08 [±0.13]; *Z* = 0.04, *p* = 0.68).

In the Replay task, observers also gave a confidence rating after each Type-I response. The rating ranged from 1 to 4 and reflected observers' confidence that they had made a correct response (1 corresponding to low confidence/guessing, and 4 high confidence/certain correct). These ratings were used to compute observers' Type-II sensitivity using meta-*d′*, as has become common practice in metacognitive research[28]. To account for different Type-I sensitivity, meta-*d′* is divided by *d′* to give metacognitive efficiency, or Type-II efficiency as we will call it here. The average Type-II efficiency was 0.75 [±0.096], more details on this analysis are available in the Supplementary Note 2 and Supplementary Fig. 2.

Observers' behavioural responses were determined by the orientations they were presented with on each trial, and by their internal processing of these samples, which is affected by several sources of suboptimalities. These suboptimalities were quantified using computational modelling, based on the work of Drugo-witsch et al.[22]. The model describes the observer's choice on each trial as an accumulation of evidence with each additional sample, as shown in Fig. 2a. The evidence accumulated with each sample was calculated as the difference in the log probabilities of the orientation given each category (blue vs. orange), which is the optimal evidence given by a Bayesian observer. This optimal evidence was disrupted by two sources of suboptimalities that impair the observer's Type-I sensitivity: inference noise (disrupting the accurate representation of decision evidence) and a temporal bias (weighting early and late evidence differently), parameterised by σ and α, respectively (see Methods for details). These parameters were adjusted for each observer to best describe their behavioural choices. In order to achieve the performance targets, the observer imposes a bound on the accumulated evidence. To maintain a constant probability of a correct response, the ideal bound decreases with the number of samples—i.e., a collapsing bound (see Methods for more details). That is, as additional samples are accumulated, the observer requires less evidence per sample on average for the same probability of a correct response. We found that observers adjusted the rate of decline of the bound function over the number of samples (parameterised by λ) to accommodate for the different target performance conditions. On average, λ increased from 2.17 [±0.38] to 3.55 [±0.53] to 4.38 [±0.69] and this alone was sufficient to explain the differences between conditions in

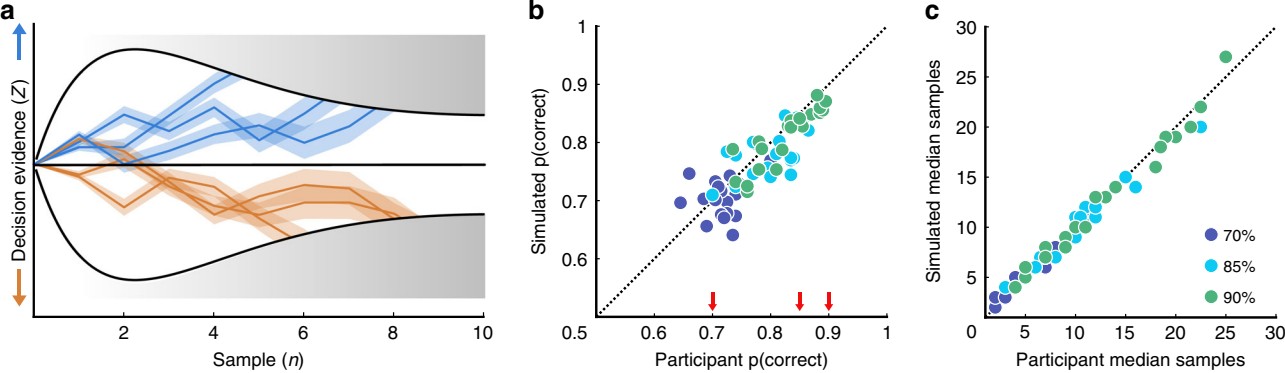

**Fig. 2 Computational model and simulations. a** On each trial, the observer sums the decision evidence ($z$) of each sample, $n$, which is the log-probability of the orientation, given the distribution ($\ell_n$), corrupted by additive i.i.d. Gaussian noise ($\varepsilon_n$, with standard deviation $\sigma$), and weighted according to the observer's leak ($v_n$, an exponential function of $\alpha$). Three example trials are shown from each distribution (blue and orange traces, with the shading representing the variability due to noise). Evidence accumulation stops once the evidence reaches the bound, in this case a collapsing bound defined by $\Lambda_n$, shown by the black curves. **b** Participant proportion correct against the simulated data based on the fitted parameters of the computational model, for each participant in each condition of the Stopping task (70%, 85% and 90% correct in blue, cyan and green, respectively). The diagonal shows equality. Red arrows mark the performance targets. **c** The same as **b** for the median number of samples the observer chose to respond to.

Fig. 2b, c. The relative placement of this bound, given the suboptimalities in their evidence accumulation, controlled how well observers were able to meet the target performance levels. The model was fit to both what and when observers responded. The ability of the model to describe observers' behaviour in each target performance condition can be appreciated by simulating behaviour using each participants' fitted parameters, as shown in Fig. 2b, c. Further details on the model parameters and fits can be found in the Methods. An indication of the model fit to behavioural performance is shown overlaid over each behavioural data figure, with open red markers corresponding to performance in simulations using the parameters fit to each observer.

This computational model was then used to examine the process of accumulating evidence for Type-I and Type-II decisions in the context of our three main questions: the relationship between efficient bound-setting for Type-I decision-making and Type-II sensitivity, the implementation of covert bounds on Type-I and Type-II evidence accumulation, and the relationship between suboptimalities in evidence accumulation for Type-I and Type-II decisions.

**The relationship between Type-I and Type-II efficiency.** Observers' ability to set and maintain appropriate bounds on Type-I evidence accumulation was measured in the Stopping task. As shown in Fig. 3a, observers were able to adjust their performance according to the target performance condition, but did not do so optimally: there was substantial over-performance in the 70% correct condition and under-performance in the 85% and 90% correct conditions. The computational model explained this change in behaviour by a change in the rate of decline of the decision bound, which also explained the increase in the number of samples observers chose to respond to (Fig. 3b). Bound efficiency was then calculated as the change in bound across conditions that the observer actually implemented, divided by the change in bound they should have implemented if they were to reach the target performance levels. This expected behaviour was obtained by simulations given the other suboptimalities affecting observers' performance (see Methods for more details). In this way, an observer with high bound-efficiency could over- or under-estimate their performance (poor accuracy relative to the target performance levels), though they still adjust their bounds appropriately in the face of different performance targets (good precision). Type-II efficiency was then calculated using the

confidence ratings in the Replay task (completed in a separate session to the Stopping task). As a check, Fig. 3c shows observers were using their ratings appropriately, as proportion correct increases with increasing confidence, and reflected increased decision evidence rather than number of samples per se (which could have been used as a proxy for confidence).

There was a strong, significant relationship between bound efficiency and Type-II efficiency, shown in Fig. 3d, found by assessing the slope of the line that minimises the perpendicular distance from the data ($y = 1.41\times + 0.25$; $p = 0.004$, using non-parametric bootstrapping; $y = 1.5\times + 0.2$, $p = 0.026$ with two outliers removed). Importantly, the bound efficiency was not found to relate to observers' Type-I sensitivity in the Replay task ($p = 0.64$), nor was Type-II efficiency related to the parameters contributing to Type-I sensitivity in the Stopping task (inference noise ($\sigma$); $p = 0.51$; and temporal biases ($\alpha$) $p = 0.17$). This indicates that the relationship between the bound efficiency and Type-II efficiency did not arise from other, potentially confounding computational parameters underlying Type-I and Type-II sensitivity.

**Covert bounds on Type-I evidence accumulation.** In the Replay task, the additional samples in the More condition should have driven increased performance compared to the Same condition, but from the statistical analysis reported in preliminary analysis, this was not the case (Fig. 4a). This lack of improvement has two possible explanations. First, there could be a performance limit, due to the suboptimalities in evidence accumulation parameterised by $\sigma$ (inference noise) and $\alpha$ (inference leak). Second, and more interestingly, observers could be employing a covert bound on evidence accumulation, where observers do not accumulate evidence beyond this bound, even when additional evidence is available. To compare these hypotheses, we fit two models, one with and one without a covert bound, fixing the suboptimalities to those fit to the Same and Less condition. There was a significant improvement in model fit with a covert bound relative to the model without a covert bound, assessed using a five-fold cross validation (mean relative increase = 0.047, $Z = 2.46$, $p_{bonf*3} = 0.041$; as shown in the left-most bar of Fig. 4b), suggesting that the employment of a covert bound is a better explanation of the behavioural responses than the suboptimalities in evidence accumulation alone. This model comparison included two other models in which a covert bound was fit to Type-II

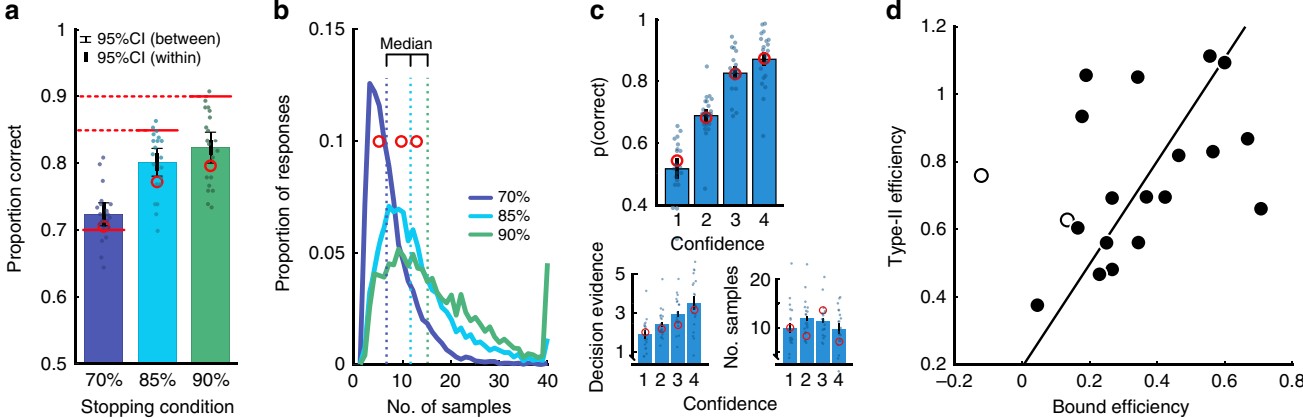

**Fig. 3 The relationship between Type-I bound efficiency and Type-II efficiency. a** Average proportion correct in each condition of the Stopping task. The horizontal red lines subtend the target performance level in each condition. Error bars show 95% within-subjects (thick) and between-subjects (thin) confidence intervals $n = 20$. The red circles show the predicted performance based on the fitted parameters of the model. **b** Average proportion of trials by the number of samples the observers chose to respond to in each condition of the Stopping task. Vertical dashed lines show the average median number of samples. Red circles show the predicted median number of samples based on the fitted parameters of the model. **c** Proportion correct (top), decision evidence (bottom left) and number of samples (bottom right) by confidence rating in the Replay task. Error bars show 95% within-subjects confidence intervals ($n = 20$), and red circles show the predicted performance at each confidence level based on the fitted parameters of the model. **d** Each observer's bound efficiency (x-axis) by their Type-II efficiency (y-axis) and the line minimising the perpendicular distance from these points, here shown excluding two outlier participants (open circles) whose optimal bound was estimated using maximum performance rather than 85% correct ($y = 1.5 \times + 0.2$, p-value for slope = 0.026, one-sided, based on bootstrapping, with outlier participants removed).

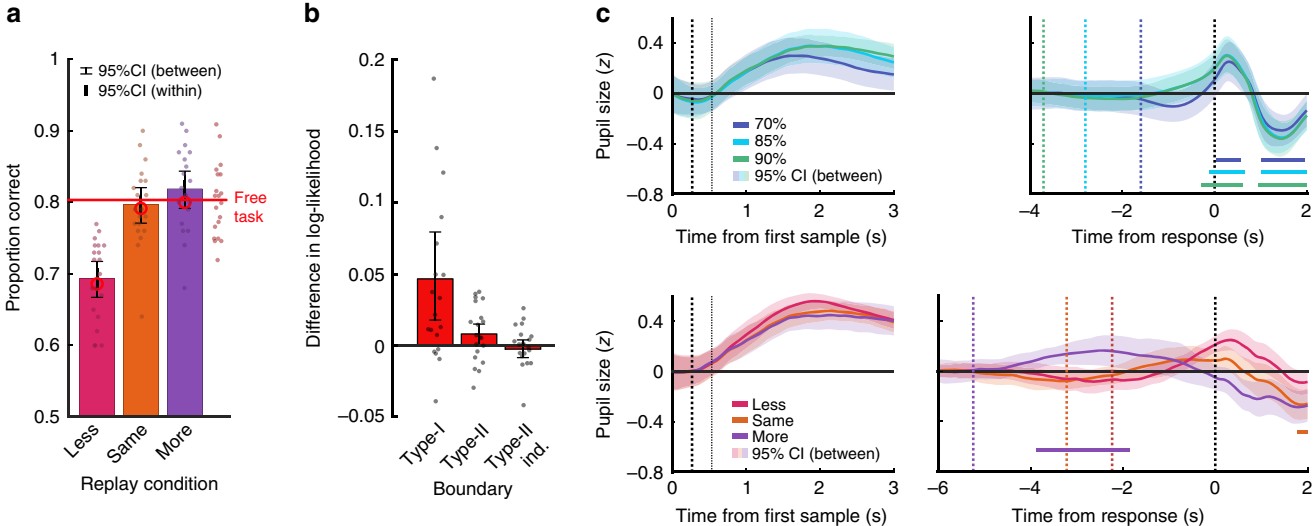

**Fig. 4 Evidence for a covert bound on Type-I evidence accumulation. a** Average proportion correct in each condition of the Replay task. In the Same condition (orange), observers were shown the same number of samples as they had chosen to respond to in the Free task, in the Less condition (magenta) they were shown two fewer samples, and in the More condition (purple), four additional samples. The red horizontal line shows the average proportion correct in the Free task, where the exact same trials were shown (with the exception of the condition manipulation). Error bars show 95% within-subject (thick) and between-subject (thin) confidence intervals ($n = 20$). Open red circles show the model predictions based on simulations of the bounded model. **b** Difference in log-likelihood (relative improvement in fit) between the model fit with a covert boundary compared to the model without. The bars show the improvement in the fit to Type-I responses with a bound on Type-I evidence accumulation (labelled Type-I), the improvement in the fit to Type-II responses when the Type-II is bounded at the same time as the Type-I evidence (labelled Type-II), and the improvement in the fit to Type-II responses when an independent bound is fit to the accumulation of Type-II evidence (labelled Type-II ind.). Error bars show 95% between-subject confidence intervals $n = 20$. **c** Standardised pupil size in the Stopping task (top) and Replay task (bottom), averaged across time-windows aligned to the start of each trial (left) and to the response (right). Shading shows 95% between-subjects confidence intervals ($n = 19$). Trial-start aligned data were baselined to the average at time 0, response-aligned data were baselined to the average prior to trial start ($-4$ to $-3$ s in the Stopping task, and $-6$ to $-5$ s in the Replay task). Black vertical lines in the trial-start plots show the timing of the 2nd and 3rd samples. Vertical lines in the response plots show the average median timing of the start of the trial. Differences from baseline in the response-aligned epochs were tested using two-sided Wilcoxon sign rank tests, with significant clusters shown with horizontal lines (corrected $p < 0.05$).

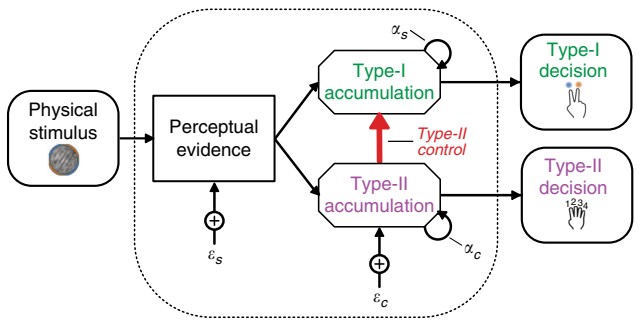

**Fig. 5 The accumulation of evidence for Type-I and Type-II decisions.** The dashed box encloses the hidden decision processes, which determine the relationship between the observable variables (the physical input and behavioural output). Information from the physical stimulus is transformed into decision evidence, which is accumulated for the Type-I decision with additive Gaussian noise ($\varepsilon_s$) and weighted according to the leak ($\alpha_s$). Evidence accumulated for the Type-II decision incurs additional noise ($\varepsilon_c$) and a separate leak ($\alpha_c$). Type-II control is exerted on the Type-I evidence accumulation process, depicted by the red arrow, where the accumulated evidence is sent to the decision output once the boundary is reached, based on the Type-II evidence. The Type-II evidence may continue to accumulate even after the boundary has been reached.

evidence accumulation (more details below) of which neither model showed significant evidence of an improvement in fit (for a hard bound, the same as Type-I; $Z = 1.79$, $p = 0.073$; nor an independent bound $Z = 0.11$, $p = 0.91$). Further details on reflexive vs. absorbing[29] covert bound comparisons can be found in the Supplementary Information (Note 5 and Fig. 7).

There was a clear effect of crossing the decision bound evident in observers' pupil dilation, which provided direct support for the modelling results. In all three conditions of the Stopping task, there was a phasic increase in pupil size beginning immediately prior to the response, peaking immediately after the response. Based on a cluster-level analysis[30] of Wilcoxon sign rank tests against baseline, this increase became significantly greater than baseline from 0.28 s prior to the response ($p_{bonf*3} < 0.002$ in each condition). In contrast, in the Replay task there were substantial between-condition differences. In the Same and the Less conditions the increase relative to baseline did not survive cluster-level comparisons ($p = 0.064$ in the Same condition, and $p = 0.281$ in the Less condition; uncorrected), but in the More condition pupil size was significantly greater than baseline from $-3.92$ to $-1.86$ s relative to the response ($p_{bonf*3} = 0.018$). Pupil size was also found to peak earlier in the More condition ($M = -2.46$ s) relative to the Less condition ($M = -2.8$ s) (using jackknife resampling $t(17) = -13.06$, $p_{bonf*3} < 0.001$) though neither peak was significantly different from the Same condition ($p > 0.1$). The difference in the median number of samples shown in the More and the Same condition was about 8. If the difference in the peaks were due to the different number of samples, there would also be a difference between conditions in the Stopping task, where there was also a difference of about 8 samples between the 70% and 90% target performance conditions (Fig. 4c).

## Suboptimalities in Type-I and Type-II evidence accumulation.
Confidence ratings were modelled by placing criteria on the accumulated evidence such that the confidence rating was given based on the position of the evidence relative to these criteria (further details in the Methods). The ideal Type-II observer is defined as the one that is using the exact same evidence as for the Type-I decision. If this were the case, there would be no systematic difference in the parameters fit to describe both Type-I

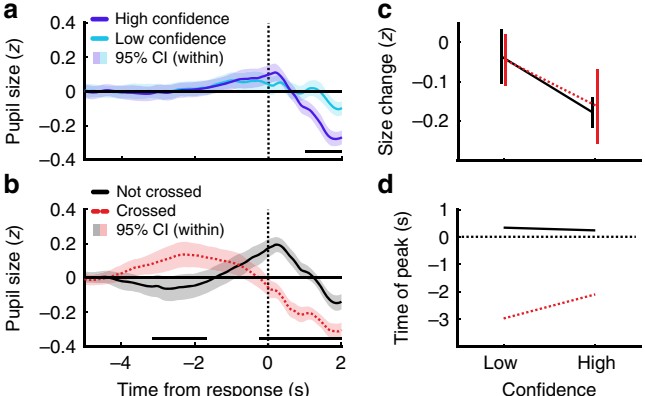

**Fig. 6 Pupil response to confidence and boundary crossing.** Average standardised pupil size in the Replay task across response-aligned (dashed vertical black line) time-windows, baselined to the average over $-5$ to $-4$ s. Shaded error bars show the 95% within-subjects confidence intervals. Differences between these lines were assessed using Wilcoxon sign rank tests, with significant clusters shown in the horizontal lines at the bottom of each plot. **a** Trials separated by confidence rating (high = 3 and 4, low = 1 and 2), with the black horizontal line showing significant differences. **b** Trials separated by whether it was likely that the observer's covert bound was crossed, based on the fitted parameters of the computational model, with the black horizontal line showing significant differences. **c** Average change in pupil size from the time of the response to 1 s after the response for low- and high-confidence trials within crossed (red dashed) and not crossed (black) trials. Error bars show 95% within-subjects CI. **d** Time of peak pupil size in the low- and high-confidence trials within crossed (red dashed) and not crossed (black) trials.

and Type-II responses, compared to when only Type-I responses were fit. On the contrary, we found a significant increase in inference noise ($\sigma$; Wilcoxon sign rank test $Z = -3.81$, $p_{bonf*2} < 0.001$), and significant decrease in temporal bias (increase in $\alpha$ toward 1; $Z = -3.62$, $p_{bonf*2} < 0.001$). If the Type-I and Type-II evidence accumulation processes were entirely independent, then there should be no significant correlation between the sub-optimalities affecting each separate accumulation process. On the contrary, we found both $\sigma$ and $\alpha$ to correlate when allowing model parameters to vary independently across the Type-I and Type-II evidence accumulation processes ($\sigma$, $\tau = 0.66$, $p_{bonf*4} < 0.001$; $\alpha$, $\tau = 0.57$, $p_{bonf*4} = 0.001$) while the Type-II parameters remained significantly increased relative to the Type-I parameters ($\sigma$, $Z = -3.92$, $p_{bonf*4} < 0.001$; and an increase in $\alpha$, $Z = -3.73$, $p_{bonf*4} = 0.001$). The results are therefore consistent with a partially dissociable model, where the Type-I and Type-II accumulators receive the same noisy decision evidence, but thereafter the Type-II accumulator incurs additional noise. This additional noise could occur either at the accumulation stage or the decision output stage, however these partially correlated models could not be distinguished (the difference in the log-likelihoods of the models is 0.67). Figure 5 depicts the additional Type-II noise as occurring at the accumulation stage. By simulating confidence ratings using the fitted parameters, we were able to closely estimate observers' Type-II sensitivity (rho = 0.59 $p = 0.007$), furthermore, estimating the placement of Type-I bounds based on the confidence evidence provided a good estimate of bound efficiency (see Supplementary Information, Note 3 and Fig. 5, for details).

A partial dissociation between Type-I and Type-II decision processes was also evident from observers' pupil dilation. There was a significant difference in pupil size between trials rated with high confidence (ratings of 3 and 4) and trials rated with low

confidence (ratings of 1 and 2) from 1.2 s following the response (Fig. 6a, cluster-level $p = 0.004$). Comparing trials predicted (based on fitted model parameters) to have crossed observers' covert bounds to those that did not cross the covert bound also revealed a significant difference in pupil size following the response (Fig. 6b, from $-0.2$ s relative to the response, $p < 0.002$). There was also a significant difference prior to the response, reflecting the earlier peak in pupil size when the bound is crossed (from $-3$ to $-1.62$ s relative to the response, $p = 0.022$). It should be noted that these trial divisions were not completely independent: 61% of the trials in which the bound was crossed were high-confidence trials, and 67% of the low-confidence trials were trials where the bound had not been crossed. Fig. 6c, d show clearly that the difference in constriction after the response is attributable to differences in confidence, while the temporal difference in when peak pupil size occurs is attributable to boundary crossing. The differences in pupil size after the responses corresponded to differences in the speed of pupil constriction, based on the analysis of the derivative, see Supplementary Information for details (Note 6 and Fig. 9).

## Discussion

The experiment presented here was designed to clarify the relationship between the accumulation of evidence for perceptual and confidence decisions. The results suggest a far more intricate relationship than has previously been assumed. While the confidence decision relies on the same sensory evidence as the perceptual decision, this evidence undergoes additional noise and a distinct temporal bias. Furthermore, the sensory evidence accumulation process is terminated by a stricter bound, in such a way that observers may make their perceptual decision prior to accumulating all available evidence, while the confidence accumulation continues. The relationship between observers' confidence efficiency and their ability to set and maintain appropriate bounds on sensory evidence accumulation suggests a common mechanism behind the two: The Type-II system imposes bounds on Type-1 evidence accumulation. Put together, this evidence indicates that confidence decisions are not the result of some inert post-decisional process, but reflect an online control process that moderates sensory evidence accumulation.

By implementing a covert bound, observers were able to terminate the Type-I evidence accumulation process prior to the end of the trial; making their decision despite there being more evidence available that could have increased their accuracy. This was evident from the behavioural results, where performance in the More condition did not significantly increase relative to the Same condition, even though there were an additional four samples on which to base their decision. Our computational modelling suggested a covert bound significantly improved the fit of the model to behavioural responses. Observers showed a phasic increase in pupil dilation as they made their decisions (the evidence reached the decision bound), which was temporally aligned with the response in the Stopping task, but occurred much earlier in the More condition of the Replay task, supporting the behavioural and modelling results that suggest that observers were covertly committing to their Type-I decisions early. This implementation of covert bounds could help to optimise perception for efficient Type-I decision-making. However, in the Replay situation, the bounds did not improve efficiency in terms of overall time to complete the task (observers were forced to wait till the end of the trial to enter their response), but rather in terms of cognitive resources (by terminating the effortful evidence accumulation process early). The Type-I evidence accumulation system is thus not only hasty, but also indolent.

In comparison, there was no evidence that the Type-II evidence accumulation system was subject to the same premature termination: the model with covert bounds did not improve the description of observers' confidence judgements, and the confidence of the observer was not discernible from their pupil dilation until after the Type-I response. This is consistent with other findings in the literature, suggesting that observers' Type-II responses inherit noise from the Type-I process[31], and may continue to accumulate Type-II evidence even after their Type-I response with additional Type-II noise[8,16], and explains how some observers are able to show superior performance in their Type-II decisions compared to what would be expected from their Type-I decisions[32]. The additional accumulation of Type-II evidence, in combination with the additional inference noise affecting Type-II evidence accumulation, can accommodate for all levels of metacognitive performance described in the literature thus far. Moreover, the particular relationship between the sub-optimalities in Type-I and Type-II evidence accumulation, with more noise but less temporal bias in Type-II accumulation supports the characterisation of these processes as a hasty but efficient (Type-I) process moderated by a cautious but inefficient Type-II process[3]. Here, these effects were found without any explicit experimental manipulation: there was no speed-accuracy trade-off to induce the early boundary on Type-I accumulation in the Replay task, and there was no manipulation of the timings of Type-I and Type-II responses to induce additional Type-II accumulation. In other words, the observed relationship between Type-I and Type-II evidence accumulation arose from observers' intrinsic predispositions.

This partial dissociation between Type-I and Type-II evidence accumulation processes was also evident from observers' pupil dilation responses, suggesting differential activations of the noradrenergic system. The pupil response to reaching the decision bound may be associated with the phasic activity of Locus Coeruleus (LC) neurons to task relevant stimuli, the timing of which is more tightly related to the response time (though occurring prior to the response) than to the timing of the stimulus[33]. This pupil response to covert decisions has been demonstrated in previous experiments[34], as has the distinct profile we saw related to confidence[35,36]. While phasic pupil dilation was associated with crossing the decision bound (temporally dissociated from the time of the response in the Replay task), a distinct effect was seen related to confidence: faster pupil constriction following the response (possibly due to an indirect effect of confidence on task disengagement). This means that these two distinct responses visible in the pupil dilation likely correspond to distinct processes that can be measured simultaneously, perhaps corresponding to the functional differences related to tonic and phasic activity of noradrenergic neurons[26].

Despite all these differences between Type-I and Type-II evidence accumulation, there was a surprisingly strong correlation between observers' Type-II efficiency and their ability to efficiently set and maintain Type-I decision bounds. It is unlikely that this correlation emerges indirectly due to some common underlying variable, such as the observers' motivation in general, because of the lack of a relationship between observer's Type-I sensitivity (which should also vary with task commitment) in the Replay task and bound efficiency in the Stopping task. Likewise, there was no relationship between Type-II efficiency in the Replay task and the magnitude of the suboptimalities in evidence accumulation in the Stopping task. The strongest interpretation of the correlation is that there is a causal relationship between Type-II efficiency and bound efficiency: Type-II evidence is being used to set and maintain boundaries on Type-I evidence accumulation (though a test for causality is left for future research). This 'metacognitive control' has been previously suggested to operate

in a post-decisional manner: the current confidence will influence future Type-I evidence accumulation[37–39]. Our interpretation goes further by postulating that the Type-II process is acting on the Type-I evidence accumulation online, as accumulation occurs. Indeed, we were able to reproduce observers' bound efficiency by simulating bounds based on the Type-II evidence estimate of the probability of correct responses (Supplementary Fig. 5). This kind of interaction would readily explain how observers actively seek more information when they are uncertain[40], and how observers can integrate loss functions into their Type-I decisions[41]. In addition, this kind of relationship offers a mechanism by which decision bounds could be implemented at the neural level, which is supported by evidence showing activity in the dlPFC is modulated according to changes in the speed-accuracy trade-off[42–44] and according to metacognitive confidence[45,46]. However, it remains to be tested whether observers utilise their Type-II evidence for bound-setting in other contexts, in particular, in contexts where the tight relationship between Type-II evidence and bound-setting is not suggested in the task instructions.

In summary, perceptual and confidence decisions were best explained by partially dissociable, yet causally related evidence accumulation processes. This intricate relationship allows human observers to simultaneously represent sensory evidence as a categorical variable while maintaining a graded representation of the uncertainty in that variable, which can then signal the commitment to a perceptual decision. This allows for fast and efficient perceptual processing under the control of more cautious confidence evaluation. Rather than adding complexity, this characterisation of perceptual processing as concomitant with the processing of uncertainty provides a general computational framework for describing several features of human decision-making. That confidence controls perceptual decision-making explains how observers can adjust decision-making across speed-accuracy trade-offs, learn to make decisions in volatile environments, incorporate priors and loss functions into perceptual decisions, and optimise perceptual processing flexibly across instances requiring detailed scrutiny and instances requiring the integration of global cues.

## Methods

**Participants**. A total of 22 participants with normal or corrected-to-normal vision were recruited via the French Relais d'Information en Sciences de la Cognition (RISC) mailing list. The experimental protocol was approved by the Conseil en éthique pour les recherches en santé and pre-registered on the Open Science Framework platform (https://osf.io/gy2t3/), in adherence with the declaration of Helsinki, and participants gave written informed consent prior to completing the experiment. Two participants were removed from the analysis based on pre-registered criteria (performance not significantly above chance), leaving 20 participants in the full analysis as planned. One further participant was removed from the pupillometry analyses due to a technical problem with the recording.

**Apparatus and stimuli**. Stimuli were presented on a 24-inch LCD monitor (BenQ) running at 60 Hz with a resolution of $1920 \times 1080$ pixels and mean luminance 45 cd/m². Stimulus generation and presentation was controlled by MATLAB (Mathworks) and the Psychophysics toolbox[47–49], run on a Mini Mac (Apple Inc). An EyeLink 1000 infrared monocular eye-tracker system (SR Research Ltd. Ontario, Canada), running at 500 Hz on a dedicated PC, was used to monitor blinks and pupil dilation in the observer's dominant eye. Observers viewed the monitor from a distance of 60 cm, with their head supported by a chin rest. Stimuli were oriented Gabor patches subtending 4° of visual angle with a Michelson contrast of 0.2 and spatial frequency of 2 cycles/degree. Gabors were embedded in spatially filtered noise with an amplitude spectrum of $1/f^{1.25}$ and contrast of 0.15. The orientation of each Gabor was chosen from one of two Von-Mises distributions with concentration parameter $\kappa = 0.5$, and means of $-45°$ and $+45°$ from vertical (0°). An annular colour guide was drawn around each Gabor to aid participants in the visualisation of these distributions, where the red and blue RGB channels reflected the probability density of each angle in the two distributions, respectively[22]. This colour guide was present throughout the trial, in addition to a black, circular fixation point subtending 0.3° at the centre of the screen. These distributions and example stimuli are shown in Fig. 1.

**Procedure**. The task was a modified version of the weather prediction task[21,22]. On each trial the observer was presented with a sequence of stimuli and was asked to guess which of the two categories (defined by the distributions of the orientations) the stimuli were drawn from. Observers were instructed to press the left arrow key for the category with mean of $-45°$ and the right arrow key for the category with mean of $+45°$, which was described to the participants using the colour guide. The stimuli were presented at a rate of 4 Hz, with Gabors presented at maximum contrast for 150 ms, temporally bordered by a 25 ms cosine ramp, with a 50 ms inter-stimulus interval. This same basic procedure was used across three variants of the task (visually depicted in Fig. 1b), completed over two sessions of approximately one hour each. For each observer, 100 trials of 40 samples were pre-defined by randomly sampling from the orientation distributions (50 trials from each distribution) and saving the random number generator seeds for recreating the spatially filtered noise added to each Gabor. These 100 pre-defined trials were repeated over the experiment.

In the first session, observers completed the Stopping task. In this task, samples were presented to the observer until they entered their response (or until all 40 samples were shown). There were three response conditions, where observers were instructed to enter their response as soon as they thought they had reached a certain performance target (a 70%, 85%, or 90% chance of being correct). These conditions were completed in a random order over six blocks of 100 trials (two blocks of each condition). Before starting this task, observers completed a practice block, where they were first shown 20 trials of 4, 8, 12, or 16 samples with immediate feedback as to which distribution the orientations were actually drawn from. They were then asked to practise responding at each of the performance targets over 10 trials for each target. During this part of the practice, observers were given feedback as to their average percent correct performance over the 10 trials. In the Stopping task, participants were informed of their average performance over mini-blocks of 20 trials, and were symbolically awarded 10 points for achieving the performance target over the 20 trials, or 5 points for coming within 5% of the target (for achieving 5% more *or* less than the target), but 0 points otherwise.

In the second session, participants completed two tasks: the Free task and the Replay task, in that order. The Free task was the same as the Stopping task except that participants were not given performance targets, but were instead asked to respond when they 'felt ready'. There were three blocks of 100 trials in this task, and participants were not given any feedback as to their performance. In the Replay task, observers were shown a specific number of samples and could only respond after the sequence finished, which was signalled by the fixation point changing to red. After entering their response, observers were also required to give a confidence rating of how certain they were that they were correct on that trial, on a scale of 1 to 4. Unbeknownst to observers, the trials in the Replay task were actually designed based on the observers' responses in the Free task. There were three conditions; in the Same condition, observers were shown the median number of samples that they chose to respond to over the three repeats of that exact same trial in the Free task. In the Less condition, observers were shown two fewer than the minimum number of samples they had chosen to respond to on that trial in the Free task, and in the More condition they were shown four more than the maximum number of samples. For example, if the observer chose to respond after 4, 5, and 10 samples on the three repeats of one pre-defined trial, they would be presented with 2, 5, and 14 samples in the Less, Same, and More conditions of the Replay task, respectively. These conditions were randomly intermixed over three blocks of 100 trials.

**Statistics and reproducibility**. This manuscript presents a single experiment with 20 participants. This sample size was pre-registered and allows the detection of a moderate effect size of 0.68 with a power of 0.8 at an alpha level of 0.05 for standard two-sided $t$-tests. The majority of statistical comparisons were performed within-subjects using non-parametric tests, making these analyses more conservative, but robust to deviations from normality, which cannot be reliably tested in small sample sizes. All measurements were taken from distinct samples unless otherwise specified.

**Behavioural analysis**. Raw behavioural data were used to calculate proportion correct, sensitivity ($d'$), and the median sequence length (number of samples observers saw) for each participant, for each experimental condition, and across experimental blocks. We present the average proportion correct across conditions in the Results. Non-parametric within-subject statistics were applied to sensitivity ($d'$) to examine differences in performance across conditions, and to the median sequence length. We also present parametric confidence intervals on the proportion correct data. Throughout the analysis, Bonferroni corrections were applied to p-values less than 0.05 when more than one statistical test was carried out per hypothesis (with this indicated by bonf*[number of tests corrected for] in subscript), while non-significant p-values were reported uncorrected.

**Pupillometry**. Blinks were identified using the EyeLink automatic blink detection algorithm and pupil dilation was linearly interpolated using a 100 ms window before and after each blink. Additional outlier pupil dilation data points (greater than 8 SD from the mean) were also interpolated. Data were downsampled to 50 Hz by averaging over consecutive windows of 20 ms. Time-points where the observer was not fixating within 200 pix from fixation were tagged for exclusion

and trials were excluded from the analysis if there was more than 100 ms of exclusory data, or if any data were more than 3 SD from the mean (on average, 2.6% of trials in the Stopping task, and 5.3% of trials in the Replay task). Each participant's pupil size data were z-scored and epochs were taken relative to the start of each trial (0 to 3 s) and relative to the response (−6 to 2 s). Epochs were baselined by subtracting the group mean for each condition at the start of the epoch from each participant. Pupil change was calculated based on the derivative of the pupil size, which was conducted with some smoothing, by taking the differences between the sum of five time-points before and after each time-point and dividing by five times the sample rate (50 Hz). Statistical inferences were performed using a cluster-based procedure[30]. Significant clusters were found using Wilcoxon sign rank tests at each time-point (at a statistical threshold of $p < 0.05$) and comparing the sum of the z-scores in these clusters to those obtained over 3000 permutations. For tests against baseline in the response-aligned epochs, data were permuted by shuffling the response times within each condition. For tests between conditions, data were permuted by shuffling the condition labels of the trials. Cluster-based corrections corrected for tests over multiple time-points, additional Bonferroni corrections were applied to significant p-values when more than one set of tests was performed (e.g., totalling three cluster-level analyses, one for each condition separately in the Stopping and Replay tasks). As a secondary check that the results were not influenced by the pupil response to stimulus onset, the analyses were also performed on data where the pupil impulse response function to stimulus onset was removed using an autoregressive model with exogenous inputs[50], these results can be found in the Supplementary Note 6 and Supplementary Fig. 8.

**Computational modelling**. A computational model was defined based on[22]. The model takes the Bayesian optimal accumulation of sensory evidence in this task, and disrupts this process with several sources parametrically defined suboptimalities. The Bayesian optimal observer is assumed to know the category means, $\mu_1 = -\frac{\pi}{4}, \mu_2 = \frac{\pi}{4}$, and the concentration, $\kappa = 0.5$, and takes the evidence in favour of category $\psi$ ($\psi = 1$ or $\psi = 2$) based on the orientation presented on a specific trial, for a specific sample $n$, as the probability of the orientation $\theta_n$ given each category:

$$p(\theta_n|\psi) = \frac{e^{\kappa \cos(2(\theta_n - \mu_\psi))}}{\pi I_0(\kappa)} \quad (1)$$

Where $I_0(\cdot)$ is the modified Bessel function of order 0. The optimal observer then chooses the category $\psi$ with the greatest posterior probability over all samples for that trial, $T$ ($T$ varies from trial to trial), given a uniform category prior, $p(\psi) \propto \frac{1}{2}$:

$$p(\psi|\theta_{1:T}) \propto p(\theta_1, \ldots, \theta_T|\psi) = \prod_{n=1}^{T} p(\theta_n|\psi) \quad (2)$$

This is achieved by accumulating the log probabilities of each category, given each orientation presented in the sequence:

$$\ell_{n,\psi} = \log p(\theta_n|\psi) = \kappa \cos(2(\theta_n - \mu_\psi)) + \text{const.} \quad (3)$$

Given the evidence for each category is perfectly anti-correlated over the stimulus orientations, the evidence from each sample can be summarised as:

$$\ell_n = \ell_{n,1} - \ell_{n,2} = \kappa \cos(2(\theta_n - \mu_1)) - \kappa \cos(2(\theta_n - \mu_2)) \quad (4)$$

and the optimal observer sums this evidence over all samples in the trial ($T$):

$$z = \sum_{n=1}^{T} \ell_n \quad (5)$$

Such that the Bayesian optimal decision is 1 if $z > 0$ and 2 if $z \leq 0$.

This optimal decision-making was disrupted by several sources of suboptimality in order to account for each observer's behaviour. First, variability is added to the evidence accumulation process, such that independent and identically distributed (i.i.d) noise, $\varepsilon_n$, is added to each evidence sample. The noise is Gaussian distributed with zero mean, and the degree of variability parameterised by $\sigma$, the standard deviation:

$$\varepsilon_n \sim N(0, \sigma^2) \quad (6)$$

This noise represents inference noise, as it is added to the decision update as opposed to the representation of stimulus orientation. It is noted that there could be some contribution of sensory noise, where the representation of the stimulus orientation does not veridically match the sensory input. However, previous evidence[22] suggests that the contribution of sensory noise to this task is minimal (only very large values of sensory noise would contribute significantly to decision variability in this task), thus no sensory noise parameter was explicitly fitted in reported analyses.

The suboptimal observer does not accumulate evidence perfectly. Functionally, during accumulation, the current accumulated evidence is weighted by $\alpha$, before accumulating the next sample, so that when $\alpha > 1$ this creates a primacy effect and later evidence affects the decision less than the initial evidence. In contrast when $\alpha < 1$ this creates a recency effect, and the observer's decision places greater weight on the more recent evidence than the initial samples. Thus, by the end of the sequence,

the weight on each sample $n$ is equal to:

$$\nu_n = \alpha^{T-n} \quad (7)$$

Where $T$ is the total samples in that trial and $n \in [1,T]$. Altogether, the suboptimal accumulation of decision evidence takes the following form:

$$z = \sum_{n=1}^{T} (\ell_n + \varepsilon_n) \cdot \nu_n \quad (8)$$

Several additional parameters were necessary to describe when observers would respond. The optimal observer makes a decision as soon as the relative decision evidence, given the sequence length ($n$), has crossed a decision boundary, $\Lambda$. In order to maintain a constant likelihood of a correct response (as required by the task) this bound was found to decrease as sequence length increases (such that the bound represents a constant bound on proportion correct over sequence length, further details in Supplementary Note 3 and Supplementary Fig. 4):

$$\Lambda_{n+} = n \times \left(a + be^{-\frac{n}{\lambda}}\right) \quad (9)$$

For the positive decision bound (the negative bound, $\Lambda_{n-} = -\Lambda_{n+}$). The likelihood $f(n)$ of responding at sample $n$ was estimated by computing the frequencies, over 1000 samples from $\varepsilon_n$ (Monte Carlo simulation), of first times where the following inequality is verified:

$$\left| \sum_{n=1}^{N} (\ell_n + \varepsilon_n) \cdot \nu_n \right| > \Lambda_n \quad (10)$$

As we do not have access to when the decision is made, only when the response is entered, two additional parameters are used to describe the mean, $\mu_U$, and variance, $\sigma_U^2$, of the non-decision time, which is assumed to be i.i.d across trials and take the form of a Gaussian distribution. Thus, the likelihood of responding over all samples $n$ is calculated as:

$$f\prime(n) = f(n) * g(n; \mu_U, \sigma_U^2) \quad (11)$$

where $f(n)$ is the likelihood of responding at sample $n$ as above in Eq. 10, $f\prime(n)$ is the modified likelihood that takes into account a smoothing of the choices in time, and $g(n; \mu_U, \sigma_U^2)$ is the Gaussian kernel with mean $\mu_U$ and variance $\sigma_U^2$ applied at sample $n$.

**Model fitting**. First, the full model was fit to Type-I behaviour in each task and condition separately. Responses in the Stopping task and the Free task were modelled by optimising parameters to minimise the negative log-likelihood of the observer making response $r$ at sample $n$ on each trial, using a Bayesian Adaptive Direct Search[51]. As there is no known analytic solution to the likelihood function of the model, the probability of the observer making each response at each sample, given the parameters, was numerically estimated using Monte Carlo simulation. The sensitivity of this approach was tested using parameter recovery. Simulating 300 trials we found a significant correlation between the simulated and recovered parameters (using a Spearman's correlation, all $p < 10^{-5}$, more details in Supplementary Note 4, and Supplementary Fig. 6). The numerical estimation approach was also applied in the Replay task for consistency, even though when only the Type-I discrimination response was fit, the model does have an analytic solution.

The full model was then simplified using a knock-out procedure, by comparing the Bayesian Information Criterion (BIC) of the full model with the BIC of models with each parameter fixed to a neutral value in turn using Bayesian Model Selection (implemented in SPM12[52,53]), for each condition of each task. The full model contained response bias and lapse rate parameters (not described above) that could be removed as they did not significantly improve the fit (the exceedance probability of the model with response bias = 0 was xp = 0.93; and fixing lapse = 0.001, xp > 0.99). There was also little evidence for response bias from the behavioural data (mean criterion = 0.03 [±0.05]). Thus, the final Type-I model contained seven parameters for fitting both what and when observers responded, and just two parameters for fitting only the categorisation response. Next, we examined whether any of the parameters systematically varied across conditions within tasks. For the Replay task, this is described in the Results. For the Stopping task, this was used to assess which parameters observers were adjusting to control their bounds. The only parameter to significantly vary across conditions in the Stopping task was $\lambda$ (Kruskal–Wallis test $\chi^2 = 14.34$, $p_{\text{bonf}*7} = 0.006$) with all other $p > 0.1$ (uncorrected; and specifically, $b$, the other important bound parameter, showed no evidence of adjustment across conditions: $\chi^2 = 1.51$, $p = 0.47$). We then fit all conditions of the Stopping task together, with three $\lambda$, one for each condition.

**Estimating optimal bounds**. In order to calculate bound efficiency, observers' actual bound separation was divided by the optimal bound separation. The optimal bound separation was estimated by simulating observers' performance across all bounds, and taking the bounds that produced the target performance. Performance was simulated by fixing all parameters except $\lambda$ (the only parameter found to systematically adapt across target performance conditions) and producing responses to the orientations shown to observers over 1000 samples of noise from

each observers' $\varepsilon_n$. Bound efficiency was then calculated as the actual difference in observers' $\lambda$ in the 70% and 85% target performance condition, divided by the simulated difference in the $\lambda$ that would have achieved 70% and 85% correct, leaving all other parameters the same. This meant that the bound efficiency really corresponded to the ability of the observer to appropriately adjust their bound, irrespective of any absolute bias to set bounds too high or too low. The data from the 90% correct condition was not used in this calculation as the model predicted that 13 observers would have never reached 90% correct, and indeed, no observer actually reached 90% correct overall in any task of the experiment.

**Fitting Type-II responses**. To fit Type-II confidence ratings, additional criteria were required to partition the evidence for each confidence rating. We examined the absolute evidence, $|\sum_{n=1}^{N} (\ell_n + \varepsilon_n) \cdot v_n|$, that observers were exposed to in the Replay task, as a function of sequence length, for each confidence rating. We found that the evidence for each confidence rating tended to follow the same function as for the ideal bound on the Type-I decision (Eq. 9; see Supplementary Note 3 and Supplementary Fig. 3 for more details). Therefore, Type-II responses were modelled by implementing three bounds ($\Lambda_1, \Lambda_2, \Lambda_3$) as the upper limit on the evidence for each confidence rating (with the highest confidence rating having no upper limit). The three bounds were modelled with the same $a$ and $b$, but different $\lambda$. Model fitting was performed using the same method as for the Type-I behaviour, except both the Type-I and the Type-II response were fit, such that the model would respond [Type-I, Type-II]:

$$
\begin{aligned}
1,1 \quad &if\ z > 0 \quad \cap \quad |z| < \Lambda_1 \\
1,2 \quad &if\ z > 0 \quad \cap \quad \Lambda_1 < |z| < \Lambda_2 \\
1,3 \quad &if\ z > 0 \quad \cap \quad \Lambda_2 < |z| < \Lambda_3 \\
1,4 \quad &if\ z > 0 \quad \cap \quad |z| > \Lambda_3 \\
2,1 \quad &if\ z < 0 \quad \cap \quad |z| < \Lambda_1 \\
&\qquad \cdots \\
2,4 \quad &if\ z < 0 \quad \cap \quad |z| > \Lambda_3
\end{aligned}
\tag{12}
$$

First, a model was fit using the same $z$ for Type-I and Type-II responses. Entirely separate parameters (leading to independent $z$ for Type-I and Type-II responses) were fit in the parallel model, where the Type-I parameters were fixed to those fit to only the Type-I responses. In partially correlated models some parameters for the Type-II $z$ were fixed to be the same as those affecting the Type-I $z$. These models compared all combinations of fixed/varied noise and leak, and compared whether additional noise was added with each sample of evidence, or a single sample of noise irrespective of sequence length. Model comparison showed a partially correlated model, where Type-II $z$ is affected by additional noise and a different leak, best accounted for Type-I and Type-II responses: the exceedance probability of this model over the model fit using the same $z$ for Type-I and Type-II responses was xp > 0.99; the exceedance probability over the parallel model was xp > 0.999; the exceedance probability over the next best partial model (a model with leak fixed to the Type-I leak) was xp = 0.54. We then compared models in which the observer accumulates Type-II evidence over all samples and models implementing a bound on Type-II evidence accumulation (either the same bound as the Type-I bound, or an independent bound). There was no evidence for a Type-II accumulation bound (Fig. 4b); Type-II $z$ accumulated evidence across all presented samples.

If the $\lambda$ of a higher confidence bound was smaller than the $\lambda$ of a lower confidence bound, this resulted in negative likelihoods (as it is paradoxical to require less evidence for higher confidence), and the model would sometimes become stuck in a local minimum. We therefore implemented plausible lower and upper bounds on the parameters, based on initial fits to participants where the model was successfully able to apply the bounds. These plausible bounds are used by the Bayesian Adaptive Direct Search to design the initial mesh of the parameter search, and by specifying increasing but overlapping plausible bounds on the $\lambda$'s, the model was able to successfully recover sensible parameters for all participants, while not limiting the model's ability to describe the behaviour of some of the more extreme participants.

**Reporting summary**. Further information on research design is available in the Nature Research Reporting Summary linked to this article.

## Data availability

Data can be downloaded from a fork of the pre-registration on Open Science Framework (https://osf.io/c9xfr/).

## Code availability

Model fitting code can be downloaded from a fork of the pre-registration on Open Science Framework (https://osf.io/s6zfb/).

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

## Acknowledgements
This project was supported by funding from Labex (ANR-10-LABX-0087 IEC), INSERM (Inserm U960), the CNRS (CNRS UMR 8248) and in part by ANR-18-CE28-0015 grant 'VICONTE'.

## Author contributions
T.B. contributed to experimental design, pre-registration, data collection, analysis, and wrote the manuscript. V.W. contributed to experimental design, pre-registration, analysis, writing of the manuscript, and sourcing funding and provision of materials. P.M. contributed to experimental design, pre-registration, analysis, writing of the manuscript, and sourcing funding and provision of materials.

## Competing interests
The authors declare no competing interests.
