## [Peer Review File · Nature Communications]

Reviewers' Comments:

Reviewer #1:

Remarks to the Author:

Balsdon et al. investigate the relationship between decision-making and confidence judgements. Normatively, these two types of judgements should be tightly linked, because type-I judgments are more accurate when they are based on the same evidence that guides type-I judgments. Yet recent papers report dissociations between choice and confidence. The paper makes two main claims: (1) confidence controls how much perceptual evidence is accrued before making a decision, and (2) decision-makers impose covert bounds on their perceptual decision but not on their confidence decision.

Overall I found the paper timely, interesting and well written, although I think there are many issues that need to be addressed before I can recommend its publication.

Comments:

(1) In the stopping task, every 20 trials participants receive feedback about their accuracy. I believe this is problematic because it creates an incentive to implement decision policies that are not stationary. For instance, if the observer thinks she is making many correct decisions in a row, she may decide to make a few errors on purpose (or respond randomly) later in the block to compensate and reach the target performance. Observers may also change decision strategies from one block to the next, depending on the feedback they got in previous blocks. For example, if performance was too high on a given block they may decide to lower the decision bounds the next time they encounter a block that has the same target performance. These alternative strategies are problematic because the paper does not test for non-stationary decision policies. The first alternative seems particularly problematic. Observers that have higher metacognitive accuracy will more closely match the target performance than observers with poor metacognitive accuracy, because they will know better when to purposely start making errors or guessing. This would establish a link between metacognitive accuracy and task performance (or bound efficiency) that is not due to the author's preferred hypothesis (that confidence controls evidence accumulation on single trials).

(2) In the model used for the 'Stopping' task, performance targets are met by changing the height of the bound on the accumulated evidence. The shape of the bound is chosen such that accuracy is independent of time. This assumption leads to two testable predictions:

- the probability of responding correctly should be independent of the number of samples observed before making a decision (i.e., the time-dependent accuracy function should be flat),
- Correct and incorrect decisions should have the same average reaction times.

Are these predictions verified in the data?

(3) The paper claims that observers impose covert bounds on their perceptual decisions, even when the stimulus-viewing duration is controlled by the experimenter. To support this claim, it is shown that performance did not change significantly between the 'same' and 'more' conditions of the 'Replay' task. Performance was 0.8 and 0.82 respectively, the same as the performance observed in the 'Stopping' task between between the 85% and 90% conditions. In the later case, however, the difference between 0.8 and 0.82 was highly significant. Can you explain why the same difference is significant in one case and not in the other? Is it that there is more variability in performance across subjects in the 'Replay' task?

(4) I am worried that the response to sensory stimuli (and not the crossing of the bound) explains the changes in pupil size in the 'Replay' task. The bottom-left panel of Figure 4c shows that there is no

modulation of the pupil size by condition. When the pupil responses are aligned on response time (bottom-right panel), the pupil dilation for the 'More' conditions peaks earlier than for the other conditions. But this could simply be an artifact of aligning three identical time-varying functions to different points in time. Obviously, if the pupil size decreases with time, the later the event to which it is aligned the earlier the peak response will appear to be. Consistent with this interpretation, it can be seen in Figure 4c (bottom-right) that the dilation of the pupil peaks approximately 3 seconds after the start of the trial (vertical lines of the corresponding color).

Further, I do not understand why in Figure 4c there is a difference of more than 2 seconds between the vertical line indicating the median start of the 'More' trials, and the line indicating the start of the 'Same' trials. Shouldn't this difference be of around 1 second, since there are only 4 more stimuli in the 'More' than in the 'Same' condition?

(5) A strong conclusion of the paper is that 'confidence decisions are not the result of some inert post-decisional process, but reflect an online control process that moderates sensory evidence accumulation'. Why not use the fitted model to test if this 'online control process' would lead to the pattern shown in Figure 3d? My suggestion is to use the confidence model derived from the 'Replay' task to simulate the 'Stopping' task and stop the accumulation based on the online confidence estimate that the model produces. For example, for a target accuracy of 75% the decision would be stopped when the confidence is 75%. The authors could then test if the simulations replicate the relationship between meta- d' and bound setting which they show in Figure 3d.

(6) Because there is no mixture of difficulty levels in the experiment, I would have expected that flat bounds would lead to a precision that does not change over time, as in the sequential probability ratio test. Why aren't the optimal bounds flat? The supplementary materials say "A flat bound is only optimal under the assumption of constant costs in accumulating evidence, and constant evidence strength, which are not valid assumptions in our setting". I do not see what makes these assumptions invalid. Is it because of the temporal weighting?

(7) The model has a parameter (α) that controls the weight of each sample of evidence, which explains why some subjects weight more the early samples ('primacy') and others weight more the late samples ('recency'). In contrast, a more common way of explaining the patterns of 'recency' and 'primacy' is through covert bounds that end the accumulation before the end of the stimulus (eg, Kiani & Shadlen 2008), and/or leak in the accumulation that leads to recency effects (e.g., Tsetsos et. al. Front Neurosci 2012). In contrast, the current paper adopts a temporal-weighting function, which is not equivalent to a leaky integrator. For example, a temporary weighing function will affect the weight of each sample regardless of how many stimuli are shown, while the impact of the leak does depend on the number of stimuli. It would be helpful if the authors explain why they chose to model recency and primary with a temporal weighting function instead of using leak and bounds. It would also be useful to see a histogram of the value of α across participants, to know how many of them show recency and primacy.

(8) Figure S2. I don't understand how can the different curves cross each other in panel c (Participants 11 & 17). Shouldn't it be the case that the 'proportional evidence' is always higher for higher levels of confidence? (as seen for example for Participant 7).

(9) In Figure 3d, why did the authors use the "line that minimizes the perpendicular distance" instead of the correlation coefficient? Is the correlation coefficient significantly positive for the data in Figure 3d?

(10) Typo on page 11 line 26: "process in terminated" -> "process is terminated"

Reviewer #2:

Remarks to the Author:

In this manuscript Baldson and colleagues are presenting the results of a study in which they analyse the role of confidence in perceptual evidence accumulation. I really liked this study, the task manipulation is very clever and the analyses are thorough (all statistical controls and out of sample validations were performed correctly). The manuscript and figures are very clear and easy to understand.

The model they developed is simple but insightful and it is used with great care to dissect the subtleties of their behavioural results. The authors collected also pupillometry data to complement the behavioural data. These pupillometry results are in my opinion the least interesting set of data presented here but by no mean of no interest. Maybe the authors might want to discussion more (even at speculative level) on the possible role of noradrenaline (measured indirectly via pupil dilation) in modulating the circuit they proposed in fig 5

The most interesting finding is that confidence provides an online control on the accumulation of perceptual evidence that triggers first order choice response. The idea that confidence is not simply a readout of choice is not new but, as far I can tell, this work provide the first compelling evidence that this 'metacognitive' control works from the very incipit of the evidence accumulation process and exert an online control on type I accumulation. Given the high variability in metacognitive abilities between individuals it would be interesting to see if between subject variability in meta-d' account for inter-individual difference in this online control process. However since the study includes only 20 participant it might be hard to detect these between subject effects. However this is not a limitation of the current study and I am sure that idea can picked up in future studies. I believe that this is an important study that will spark interest in the community and I am therefore very happy to recommend it for publication in Nature Communications.

One more thing: I could not find a link to the data or to the analyses scripts. I know that some of the authors share my enthusiasm for transparent and open access science and I would like to kindly nudge them to make their analyses and data available on a sharing platform like GitHub. This would certainly increase the impact of their work.

Regards,

Benedetto De Martino

Reviewer #3:

Remarks to the Author:

This is a strong and interesting manuscript in which the authors weave together well-designed experiments, computational modeling, and pupillometry to make substantive contributions to the field's growing understanding of the relationship between perceptual (type-I) and metacognitive (type-II) processing. Most provocatively, they demonstrate a correlation between efficiency in perceptual decision making and metacognitive efficiency that adds to a growing body of evidence suggesting how metacognition not only evaluates perceptual processing, but may also directly participate in and shape it (Fig 3d). They additionally provide modeling and pupillometry evidence that subjects set covert decision bounds, neglecting to consider additional stimulus evidence once a decision has been made (Fig 4). Rounding things out, they demonstrate a very clean dissociation

between pupillometric signatures of perceptual decision-making and confidence (Fig 6) and perform additional modeling analyses to flesh out a bigger-picture view of how type-I and type-II processes may be related (Fig 5).

To my mind, the most interesting aspect of this manuscript is the demonstration of a link between optimality in metacognition and optimality in perceptual decision-making, which could be interpreted as potentially being a tell-tale sign of a type-II process exerting top-down control on a type-I process. However, one consideration that significantly moderates my enthusiasm is that the 'Stopping' task requires subjects to provide a type-I response once they've reached a certain threshold of estimated probability that their response will be correct, given the evidence accumulated thus far. But this means that the very nature of the task requires subjects to yoke the type-I response ("I think the stimuli were drawn from the blue distribution") to a type-II judgment ("I will only provide my type-I response once I think I'm 80% likely to be correct"). So, perhaps it is not so surprising that the subjects who are poor at distinguishing correct from incorrect responses with a 4-point confidence rating scale (poor type-II efficiency) are also poor at producing an absolute probability estimate of how likely they are to be correct (poor bound efficiency). More broadly, I worry that the type-I / type-II correlation could possibly be a contrived consequence of the task design and instructions in the 'Stopping' task, rather than reflecting a more general relationship that can be observed in more naturalistic or open-ended task settings that do not so explicitly encourage subjects to yoke type-I responses to type-II judgments in this way. I do not think this is a fatal objection or a completely trivializing observation, but it should be addressed somewhere in the Discussion.

Although I do not count this against the current manuscript, some gaps in the logic of the overall argument that could be filled out in future research are: (1) the computational modeling does not incorporate a mechanism that reflects a possible direct interaction between type-II efficiency and the setting of type-I decision bounds; (2) ideally, there would be a way to test for a causal relationship between type-II processing and type-I decision making. It may be worth discussing these points briefly in the Discussion as areas for future work to address.

Overall I think this is a strong, well-composed, thought-provoking work that provides a valuable stepping stone in the literature for future studies to more closely investigate the important and fascinating topic of possible direct and online interactions between type-I and type-II processes.

Below I provide a more detailed set of smaller points pertaining to details of the analyses, arguments, and presentation.

Results

- It would be helpful to mention in the Results that the statistics are for $n=20$ subjects. This is a basic and important enough detail that one shouldn't have to refer to the Methods to find it.
- The authors report confidence intervals for proportion correct, but seem to perform all statistical tests on d' , which caused me some initial confusion. Given that the statistical analyses were performed on d' , it would seem more pertinent to have confidence intervals reported for d' instead of, or in addition to, proportion correct. It would also be relevant to know if observers exhibited any response bias, which could influence the proportion correct data.
- It is unclear why the authors choose to use the non-parametric Wilcoxon sign rank test when comparing d' across conditions, rather than a parametric t-test which would likely offer greater statistical power. If the concern is about how the assumptions of the t-test might be inaccurate for the sampling distribution of d' , the authors could refer to the following work to more accurately perform statistical inference on d' : Miller, J.O. (1996). The sampling distribution of d' . Perception &

Psychophysics, 58, 65–72.

- Perhaps related to the above, I found it odd that the reported confidence intervals for proportion correct in the 85% and 90% conditions of the Stopping task (0.80 ± 0.021 and 0.82 ± 0.023) were nearly identical to those of the 'Free' and 'More' tasks (0.80 ± 0.025 and 0.82 ± 0.025), and yet the results of the Wilcoxon signed rank tests yielded a significant difference in the former case ($Z=2.35$, $p=0.037$) but was nowhere close to significant in the latter ($Z=0.04$, $p=0.68$). I realize the statistical tests were performed on d' rather than proportion correct, and that the 'Free' and 'More' data were slightly noisier, but the stark difference in the Wilcoxon tests is still somewhat surprising. Do these two statistical comparisons look more similar if they are assessed with a parametric test rather than non-parametric?

This is not an entirely trivial point since the authors interpret the lack of statistically significant difference in task performance between the 'Free' and 'More' conditions as one piece of evidence for the claim that subjects set covert decision bounds that terminate evidence accumulation prior to perceiving all the additional stimuli in the 'More' condition. Even if an alternative test did reveal a significant difference between the 'Free' and 'More' conditions, I don't think it would make all that much of a difference to the authors' overall argument, but it would nonetheless be informative to get clearer on all this.

- p. 5: "In order to achieve the performance targets, the observer imposes a bound on the accumulated evidence, which decreases with the number of stimuli— i.e., a 'collapsing' bound (see Methods for more details)."

While reading the Results, I had in the back of my mind the question of why collapsing rather than fixed bounds were used. The justification is sensible-- simulations show that ideal observer must use these bounds in order to yield the target task performance. It would be helpful to briefly state this justification in the Results, referring the reader to the Methods for more details, rather than burying the first mention of this point much deeper in the manuscript.

- The result plotted in Fig 3d as described in the figure legend is different from that reported in the main text of the Results, apparently due to omitting two subjects from analysis in the former but not the latter. Why the discrepancy? It is fine to only show the single result in Fig 3d, but both results should be described in the Results, with an expanded justification for omitting two subjects in the modified analysis-- the justification provided in the figure legend is not entirely clear.

Additionally, presumably the p-value reported in the figure legend pertains to the statistical significance of the slope of the fitted equation, is that right? Please clarify in the legend.

- p. 7: "We found the suboptimalities in evidence accumulation were insufficient to explain the lack of increase in performance in the 'More' condition relative to the 'Same' condition (see Supplementary Materials for details)."

I find the logic of the argument referred to in the Supplementary Materials here puzzling. It seems to run as follows: (1) when we fit parameters separately for the 'Same' and 'More' conditions, the 'More' condition had more noise and less leak; (2) it is unrealistic for these parameter values to be different, given that the 'Same' and 'More' conditions were randomly interleaved and subjects could not know ahead of time which trial was from which condition; (3) therefore differences in noise and leak cannot explain why task performance was not much better in 'More' than in 'Same'.

I think (2) is a valid point, but pointing out (2) means performing step (1) is superfluous.

Furthermore, the conclusion (3) does not seem to address the main point, which is the possibility that existing levels of noise and leak in the 'Same' condition might fundamentally constrain how much 4 extra samples in the 'More' condition can improve task performance.

I think a far more informative analysis would be as follows. First, state point (2), that it is a priori unreasonable to fit different noise and leak parameters for the 'Same' and 'More' conditions. Use this as justification for fitting noise and leak only to the 'Same' condition, in both the covert-bound and no-bound versions of the model. Using the parameters from this fit, perform simulations to see how much each model predicts task performance should improve, on average, by presenting 4 additional evidence samples, and assess how this compares to the actual data. If the no-bound model predicts that task performance should improve substantially more than it actually does in the data, then you are justified in concluding that "the suboptimalities in evidence accumulation (i.e. the noise and leak fitted to the 'Same' condition) are insufficient to explain the lack of increase in performance in the 'More' condition relative to the same condition." However, if the no-bound model predicts that the increase in task performance is numerically similar to what is actually observed, then it would seem that the suboptimalities in evidence accumulation (i.e. the noise and leak fitted to the 'Same' condition) ARE indeed sufficient to explain the lack of task performance increase.

Of course, either way, the model comparison analysis in Fig 4b is still good evidence that even if the no-bound model is *plausible* to explain the data, the covert-bound model is still *better*.

- Fig 4b, please label the y-axis more clearly. Presumably what is being quantified here is something like change in log likelihood?

- In Fig 5, leak is illustrated as α but described in the legend as ν . To this point in the manuscript, I believe ν has only been mentioned briefly in the legend of Fig 2, and the relationship between ν and α is not discussed at length til the Methods. The discrepancy between the presentation here using α in the figure and ν in the legend, is potentially confusing if one has not yet read the methods. It would be helpful to briefly clarify the relationship between α and ν in the figure legend.

- p. 9: "...we found both σ and α to correlate when allowing model parameters to vary independently across the Type-I and Type-II evidence accumulation processes..."

This is the only mention in the manuscript of a modeling approach that directly reflects the model structure of Fig 5. However, the given explanation of the modeling approach is brief and not entirely clear, and it is not described more fully in the Methods. Presumably what the authors mean is that they modeled two independent evidence accumulation processes, call them z_I and z_{II} , such that type-I response was determined by z_I crossing the covert bound, whereas confidence was determined by where z_{II} fell relative to the three confidence bounds. Is that right? It is also not clear when the readout of confidence from z_{II} occurred in this modeling approach. Did it occur at the same time as z_I crossed the decision bound, or did it occur after the last presentation of the sample? It seems appropriate to give a fuller characterization of this modeling approach in the Methods.

- It may be worth noting that this model in which type-II decisions inherit noise from type-I processes, while receiving additional type-II noise is consistent with other models put forth in the literature, e.g. Mueller & Weidemann, 2008; Jang, Wallsten, & Huber, 2012; De Martino, Fleming, Garrett, & Dolan, 2013; Maniscalco & Lau, 2016; Rahnev, Nee, Riddle, Larson, & D'Esposito, 2016; Berg & Ma, 2016; Bang, Shekhar, & Rahnev 2018.

- p. 10: "... (Figure 6c, from -0.2 s relative to the response, $p < 0.002$)"

This is referring to Fig 6b, not 6c.

- It would be interesting to briefly note somewhere in the manuscript, perhaps in the Supplementary Materials, what values of meta-d' and meta-d'/d' were produced by the best/most comprehensive model when fitted to the Replay data. Were they at all close to the empirical values?

Methods

- There is some ambiguity in how the variable T is defined that should be cleared up. On p. 17, following eq. 1, T is identified with the number of samples for a given trial, such that T varies from trial to trial. This makes it sound as if T is the total number of samples that end up being displayed on a given trial.

However, when T appears in the equations (eqs. 2, 5, 7, 8), it seems to denote not "the eventual number of samples shown during this trial," but rather "the number of samples that have been shown *up to and including the current sample.*" This would mean that T varies not only *between* trials, but also across time *within* a trial. This is especially important for understanding eq. 7, since a value of T that changes over time to indicate the total number of samples seen so far would be needed to capture the intended "leak" effect in ν_n . For instance, at the first sample of a trial, $T=1$ and the weight placed on the first sample $\nu_1 = \alpha^0 = 1$. By the tenth sample of a trial, $T=10$ and now $\nu_1 = \alpha^9$, a much lower value than 1 when $\alpha < 1$.

- p 18: "Functionally, the observer weights the current accumulated evidence by alpha, so that..."

This seems incorrectly stated, as according to eq 8, evidence sample n is weighted by $\nu_n = \alpha^{(T-n)}$, not alpha. Furthermore, if T denotes the current sample, then the weight placed on the current sample (which is presumably what "current accumulated evidence" in the quoted sentence is referring to) would always be $\nu_T = \alpha^{(T-T)} = 1$. A clearer way of phrasing this might be something like "The weight placed on evidence sample n is weighted by $\nu_n = \alpha^{(T-n)}$, where T is the current sample, so that when $\alpha > 1$ this creates a primacy effect and later evidence affects the decision less than the initial evidence, and when $\alpha < 1$ this creates a recency effect, and the observer's decision places greater weight on the more recent evidence than the initial samples."

- In the Model Fitting section, a formal equation for the log-likelihood that was maximized by the model fitting procedure should be provided.

- p. 19: "The probability of the observer making each response at each stimulus, given the parameters, was numerically estimated by taking 1000 samples from ϵ_n , ..."

This is unclear; please explain this procedure sufficiently clearly and explicitly that someone wishing to reproduce the analysis method could easily and unambiguously do so. There are other parts of the methods that make use of similar language regarding "taking 1000 samples from ϵ_n " that should be similarly clarified.

- p. 20: "We examined the ideal evidence, ..."

The mathematical expression listed directly following this quote is not the ideal evidence, as it includes the error and temporal bias terms ϵ_n and ν_n . As the authors state in the supplementary material, they define the ideal evidence as evidence uncorrupted by decision noise and temporal bias.

- In the methods, the authors state that the three bounds for defining the four levels of confidence were modeled as exponential functions with the same a and b parameters, but different λ . However, in Fig S2c, the exponentials fit to confidence data for the sample subjects appear to be fits that allow a , b , and λ to vary. It would be helpful to add panels in Fig S2 to show examples of the three fitted λ bounds superimposed on proportional evidence as a function of # of stimuli and confidence level, as this would give a better sense of how the modeling approach relates to the actual data.

- The authors also mention that if the λ value of a higher confidence bound is smaller than that of a lower confidence bound, this can cause issues for fitting the data. Presumably this is because, given that a and b are fixed, smaller λ for the higher confidence bound would entail smaller proportional evidence at a given level of # of stimuli, which would entail the paradoxical result that less evidence would be required in order to yield the higher confidence rating than for the lower confidence rating-- is that right? It might be helpful to make this point more explicitly in the text. A simple way to avoid this issue would be to enforce a constraint on the λ parameters such that $\lambda_1 \leq \lambda_2 \leq \lambda_3$. Was that constraint actually enforced? It is not clear given the way the methods text is currently written.

Minor points

- p. 1: "The Type-II decision estimates the likelihood of the Type-I decision, given the accumulated evidence."

The term "likelihood" is ambiguous here. Presumably the authors mean to refer to the likelihood that the Type-I decision is correct.

- eq 2: in the $p(\theta_1, \dots, \theta_{T_n} | \psi)$ term, the subscript on the last θ should be T rather than T_n .

- eq 9: " λ_n " should be " Λ_n "

- p. 18: "The likelihood $f(n)$ of responding at sample n was estimated by computing the frequencies, over 1000 particles, ..."

Unclear what "particles" means here.

- p. 18: "the mean, T , and variance, $(\sigma_T)^2$, of the non-decision time..."

T should not be used as the mean of the non-decision time since it is previously used to mean the current sample in a trial, and/or the total eventual number of samples in a trial. However, it looks as though this usage of T is replaced with U later on. If that is right, then the T s that appear in the quoted sentence should be replaced with U s.

- p. 18: " $g(n; U, \sigma^2)$ is the Gaussian kernel..."

σ^2 is missing the U subscript.

signed,
Brian Maniscalco

We thank the three reviewers for providing in-depth and insightful comments on our manuscript. We have tried to make the best use of their reviews to strengthen the main findings reported in the manuscript, and clarify their presentation. We have separated the points to be addressed, and labelled them by reviewer and comment (e.g., **R1C02** refers to the second comment of reviewer #1). In the revised manuscript and its supplementary materials document, we have commented each relevant change with the label in the response. In our response to reviewers' comments below, we also provide the page and line numbers (e.g., **P19L612** refers to page 19, line 612, according to the continuous line numbering of the manuscript), so that our revisions can also be easily found even in the cleaned revised manuscript. Text written by reviewers is in dark blue and indented.

Reviewer #1

Balsdon et al. investigate the relationship between decision-making and confidence judgements. Normatively, these two types of judgements should be tightly linked, because type-I judgments are more accurate when they are based on the same evidence that guides type-I judgments. Yet recent papers report dissociations between choice and confidence. The paper makes two main claims: (1) confidence controls how much perceptual evidence is accrued before making a decision, and (2) decision-makers impose covert bounds on their perceptual decision but not on their confidence decision.

Overall I found the paper timely, interesting and well written, although I think there are many issues that need to be addressed before I can recommend its publication.

Comments:

R1C01 (1) In the stopping task, every 20 trials participants receive feedback about their accuracy. I believe this is problematic because it creates an incentive to implement decision policies that are not stationary. For instance, if the observer thinks she is making many correct decisions in a row, she may decide to make a few errors on purpose (or respond randomly) later in the block to compensate and reach the target performance. Observers may also change decision strategies from one block to the next, depending on the feedback they got in previous blocks. For example, if performance was too high on a given block they may decide to lower the decision bounds the next time they encounter a block that has the same target performance. These alternative strategies are problematic because the paper does not test for non-stationary decision policies. The first alternative seems particularly problematic. Observers that have higher metacognitive accuracy will more closely match the target performance than observers with poor metacognitive accuracy, because they will know better when to purposely start making errors or guessing. This would establish a link between metacognitive accuracy and task performance (or bound

efficiency) that is not due to the author's preferred hypothesis (that confidence controls evidence accumulation on single trials).

The reviewer has framed the issue of possible non-stationarities in participants' decision policies very astutely. We agree that participants could in theory rely on such strategies to adjust their ongoing performance to target levels. We have thus performed additional fine-grained temporal analyses of the behavioural data, and added an additional section to the supplementary materials document to address this concern (commented as **R1C01**; **P01L002**). If observers were implementing a non-stationary policy, then they should show some sign of non-stationary behaviour: In particular, we would expect to find lower performance and fewer samples being observed towards the end of the block. We thus calculated performance and mean number of samples as a function of trial position in each sub-block of 20 trials. As can be seen below, there is no evidence for non-stationary decision policies (**Supplementary Figure S1a**; note that the behavioural measures are more noisy because there are only 10 trials in each estimate). Splitting sub-blocks in half, we can also see little evidence for a systematic change in behaviour towards the end of each sub-block. Further, as the reviewer suggests in **R1C02**, an observer with a flat bound on proportion correct should show equal probability of correct and incorrect responses independent of the number of samples accumulated. This analysis has also been added to the supplementary, where no observer shows a strong tendency of accumulating fewer samples for incorrect responses (as would be predicted with a non-stationary decision policy). The new section of the supplementary materials document is quoted below:

"Our model assumed that, in the stopping task, observers entered their response when they thought they had accumulated enough evidence to meet the target performance on each trial, as they were instructed to do. We gave observers feedback on their average performance over sub-blocks of 20 trials. Observers could have tried to meet the target performance using a different strategy, for example, aiming for 100% accuracy on the first 14 trials, and then 0% accuracy on the next six trials (to achieve 70% correct), or aiming for 100% accuracy on the first 10 trials and then guessing (50% accuracy) on the next 10 (to achieve 75% correct). Or more simply, observers who felt as though they performed too well throughout the majority of the sub-block could start making errors, or responding after very few samples, in order to lower their accuracy. If this were the case, there would be a difference in accuracy across the block, and possibly a difference in the number of samples. There was no evidence for this, as shown in **Figure S1a**. We instructed observers to respond when they thought they had a certain probability of a correct response on each trial, in other words, employ a flat bound on proportion correct. A flat bound on proportion correct would mean that the probability of a correct response should be independent of the number of samples: there should be no difference in the distribution of correct and incorrect responses over the number of samples. **Figure S1b** (top) shows these distributions are indeed closely matched. Further, an observer who employs a non-stationary decision policy, where they purposefully make some guesses, would probably show fewer samples for incorrect responses on average (a guess can be made after a single sample, and is more likely to result in an incorrect response). No observer showed a substantial change in the number of

stimuli for correct and error responses (**Figure S1b**, bottom), and at the group level there was no significant difference in the number of samples for correct and incorrect responses, except in the 85% correct condition, where there was actually an increase in the average number of samples for incorrect responses (of approximately one sample on average, $p_{\text{bonf}^*3} = 0.01$).

Figure S1. Behaviour within sub-blocks. **a)** Top left: proportion correct by trial number in sub-block. Fine lines show individual subjects (10 trials per data-point), thick lines show average across subjects. Top right: proportion correct in the 1st and 2nd half of the sub-blocks. At the extreme, to obtain 70% correct using a deviant strategy, an observer would need to be at 100% in the first half, and 50% in the second half. Bottom left: average number of samples by trial in sub-block. Bottom right: number of samples by 1st and 2nd half of the sub-blocks. At the extreme, to obtain 70% correct using a deviant strategy an observer would need to look at around 15 samples in the first half, and just one sample in the second half. **b)** Top: proportion of correct responses by number of samples (solid) and proportion of incorrect responses by number of samples (dashed), colours show the different conditions in the ‘Stopping’ task. Bottom: median number of samples for correct and incorrect responses for each participant (thin lines) and averaged across participants (thick lines; correct responses shown with filled markers, incorrect with empty), for each condition in the ‘Stopping’ task.

R1C02 (2) In the model used for the ‘Stopping’ task, performance targets are met by changing the height of the bound on the accumulated evidence. The shape of the bound is chosen such that accuracy is independent of time. This assumption leads to two testable predictions:

- the probability of responding correctly should be independent of the number of samples observed before making a decision (i.e., the time-dependent accuracy function should be flat),
- Correct and incorrect decisions should have the same average reaction times.

Are these predictions verified in the data?

The reviewer is correct that the shape of the bound is chosen such that the likelihood of a correct response is indeed stable across time. Even though human observers may set their bounds imperfectly, our model predicts that they should set their bounds such that their *subjective* (estimated) likelihood of a correct response is independent of the number of samples observed before making a decision. As can be seen in **Figure 3C**, participants' confidence (i.e., their subjective likelihood of a correct response) is indeed independent of the number of samples – as predicted qualitatively by our model of the 'Stopping' task. Regarding the second prediction, we interpreted 'reaction times' as 'number of samples observed before making a decision'. We have compared the distributions of 'reaction times' between correct decisions (solid lines) and errors (dashed lines), for each target performance level of the 'Stopping' task (**Figure S1b**, copied above). As predicted by the model, correct decisions and errors were indeed associated with highly similar distributions of 'reaction times'. Correct and incorrect decisions have very similar median number of samples, with no significant within-subjects differences, except in the 85% correct condition where incorrect responses occurred on average after one additional sample. We thank the reviewer for prompting this additional test of the model.

R1C03 (3) The paper claims that observers impose covert bounds on their perceptual decisions, even when the stimulus-viewing duration is controlled by the experimenter. To support this claim, it is shown that performance did not change significantly between the 'same' and 'more' conditions of the 'Replay' task. Performance was 0.8 and 0.82 respectively, the same as the performance observed in the 'Stopping' task between between the 85% and 90% conditions. In the later case, however, the difference between 0.8 and 0.82 was highly significant. Can you explain why the same difference is significant in one case and not in the other? Is it that there is more variability in performance across subjects in the 'Replay' task?

We understand the reviewer's concern, which was also raised by reviewer #3 (**R3C06**). The nature of the comparison was not adequately described in the original manuscript. Indeed, we left an important detail to the methods section (**P15L462**, and **P15L488**), which can explain the apparent discrepancy pointed out by the reviewers. We have tried to explain this detail better in the results now, whilst maintaining the brevity required by *Nature Communications* (commented **R1C03a**, **P03L81**; **R1C03b**, **P04L129**; **R1C03c**, **P04L134**). This detail is that the same 100 pre-defined trials for each participant were repeated throughout the entire experiment. In the 'Free' task, observers completed each trial three times. Therefore, for each trial, there were three responses (3 repeats) and three 'number of samples' ('Less', 'Same', and 'More' conditions). For the 'Less' condition of the 'Replay' task, we took the *minimum* number of samples minus 2 samples. For the 'Same' condition, we took the *median* number of samples. And for the 'More' condition, we took the *maximum* number of samples plus 4 samples. For clarity, an explicit example of this procedure has been added to the methods, commented **R1C03d** on **P15L490**:

“For example, if the observer chose to respond after 4, 5, and 10 samples on the three repeats of one pre-defined trial, they would be presented with 2, 5, and 14 samples in the ‘Less’, ‘Same’, and ‘More’ conditions of the ‘Replay’ task respectively.”

The comparison of d' was not conducted across all trials of the ‘Free’ task (300), but for the 100 trials chosen for each condition (‘Less’, ‘Same’, and ‘More’) in the ‘Replay’ task. Thus, the comparison asks if precisely 4 additional samples improve d' , not if sensitivity is improved in the ‘More’ condition relative to the ‘Free’ task taken as a whole. This has been clarified and commented as **R1C03e, P05L135**:

“In the ‘Less’ condition, two fewer samples corresponded to a substantial decrease in performance within subjects ($M = 0.69 [\pm 0.025]$, Z (‘Less’ vs. ‘Free’ d') = 3.51, $p_{\text{bonf}^*3} < 0.001$) but in the ‘More’ condition, four additional samples did not significantly improve performance within subjects ($M = 0.82 [\pm 0.025]$; $Z = 0.04$, $p = 0.68$).”

Importantly, when we compare d' in the ‘More’ condition to d' calculated across all trials of the ‘Free’ task, the difference is also not significant ($Z = 1.34$, $p = 0.18$). Note, also, that because our test is a within-subject comparison, it is important to consider the standard error of the difference. In the ‘Stopping’ task, d' was on average 1.74 for an 85% target and 1.92 for a 90% target, corresponding to an average difference of 0.18 and a 95% within-subject confidence of 0.064 (yielding a significant difference). In the ‘More’ condition, d' was on average 1.85 and in the ‘Free’ task (all trials) it was on average 1.75, corresponding to an average difference of 0.095 (about half the above) and a 95% within-subject confidence of 0.068. In other words, there is a substantially smaller within-subject effect size for the difference between the ‘More’ condition and the ‘Free’ task than between the 90% and the 85% target conditions within the ‘Free’ task. We hope that the additional clarifications reported in the revised manuscript will alleviate this apparent discrepancy.

R1C04 (4) I am worried that the response to sensory stimuli (and not the crossing of the bound) explains the changes in pupil size in the ‘Replay’ task. The bottom-left panel of Figure 4c shows that there is no modulation of the pupil size by condition. When the pupil responses are aligned on response time (bottom-right panel), the pupil dilation for the ‘More’ conditions peaks earlier than for the other conditions. But this could simply be an artifact of aligning three identical time-varying functions to different points in time. Obviously, if the pupil size decreases with time, the later the event to which it is aligned the earlier the peak response will appear to be. Consistent with this interpretation, it can be seen in Figure 4c (bottom-right) that the dilation of the pupil peaks approximately 3 seconds after the start of the trial (vertical lines of the corresponding color).

The reviewer is correct that different numbers of samples during stimulus presentation has an effect on the time course of pupil dilation. The plots on the left of **Figure 4c** (copied below for ease of reference) show that the pupil *dilates* in response to stimuli and this dilation saturates in about 1.5 s (approximately 6 samples). After this time point, pupil dilation remains fairly stable for the next second (beyond which the data locked to the onset of the first sample become less interpretable, as pointed out by the reviewer, because of the different response times). In the ‘Stopping’ task (top row), the median number of samples was 6.5, 11.5, and 15 for the different target performance levels (70%, 85%, and 90%, respectively). As can be seen on the right plot, aligned to response onset, the pupil showed a transient peak at the same latency after response onset despite large differences in the number of samples seen until the response. This is consistent with our interpretation that this transient peak is aligned to decision time, not to the onset of stimulus presentation. In the ‘Replay’ task (bottom row), the median number of samples was 5.4, 10.4 and 18.5 in the ‘Less’, ‘Same’, and ‘More’ conditions, respectively – i.e., quite comparable to the differences found between target performance levels in the ‘Stopping’ task. If the difference in the peaks of pupil dilation aligned to response onset during the ‘Replay’ task was due to the effect of stimulus presentation, then there should be a similar difference in the ‘Stopping’ task. However, as emphasized above, the peaks are clearly well aligned in the ‘Stopping’ task. Therefore, the genuine effect of stimulus presentation on pupil dilation cannot alone account for the different times of pupil dilation peak found between the ‘Less’, ‘Same’ and ‘More’ conditions in the ‘Replay’ task. A statement that summarizes this line of reasoning has been added to the revised manuscript, commented as **R1C04, P08L256**:

“The difference in the median number of samples shown in the ‘More’ and the ‘Same’ condition was about 8. If the difference in the peaks were due to the different number of samples, there would also be a difference between conditions in the ‘Stopping’ task, where there was also a difference of about 8 samples between the 70% and 90% target performance conditions (**Figure 4c**).”

Figure 4. Evidence for a covert bound on Type-I evidence accumulation. a) Average proportion correct in each condition of the ‘Replay’ task. In the ‘Same’ condition, observers were shown the same number of samples as they had chosen to respond to in the ‘Free’ task, in the ‘Less’ condition they were shown two fewer samples, and in the ‘More’ condition they were shown four additional samples. The red horizontal line shows the average proportion

correct in the 'Free' task, where the exact same trials were shown (with the exception of the condition manipulation). Error bars show 95% within-subject (thick) and between-subject (thin) confidence intervals. Open red circles show the model predictions based on simulations of the bounded model **b**) Difference in log-likelihood (relative improvement in fit) between the model fit with a covert boundary compared to the model without. The bars show the improvement in the fit to Type-I responses with a bound on Type-I evidence accumulation (labelled Type-I), the improvement in the fit to Type-II responses when the Type-II evidence stops accumulating at the same time as the Type-I evidence (labelled Type-II), and the improvement in the fit to Type-II responses when an independent bound is fit to the accumulation of Type-II evidence (labelled Type-II ind.). Error bars show 95% between-subject confidence intervals **c**) Average standardised pupil size in the 'Stopping' task (top) and 'Replay' task (bottom), separated by conditions, and averaged across time-windows aligned to the start of each trial (left) and to the response (right). Shading shows 95% between-subjects confidence intervals. Trial-start aligned data were baselined to the average at time 0, response aligned data were baselined to the average prior to trial start (-4 to -3 s in the 'Stopping' task, and -6 to -5 s in the 'Replay' task). Black vertical lines in the trial-start aligned plots show the timing of the 2nd and 3rd samples. Vertical lines in the response-aligned plots show the average median timing of the start of the trial in each condition. Differences from baseline in the response-aligned epochs were tested using Wilcoxon sign rank tests, with significant clusters shown with horizontal lines.

Further, I do not understand why in Figure 4c there is a difference of more than 2 seconds between the vertical line indicating the median start of the 'More' trials, and the line indicating the start of the 'Same' trials. Shouldn't this difference be of around 1 second, since there are only 4 more stimuli in the 'More' than in the 'Same' condition?

This point is related to the apparent discrepancy raised by the reviewer, which should have been clarified above in response to **R1C03**. To summarize, the difference between the 'More' and 'Same' conditions is not *exactly* 4 samples, but *at least* 4 samples. The median difference is actually 8 samples (i.e., approximately 2 seconds), which is consistent with the observed differences in the timings of the pupil dilation peaks locked to response onset. This important detail has been added to the revised manuscript, commented **R1C04**, **P08L256** (quoted above in response to **R1C03**).

R1C05 (5) A strong conclusion of the paper is that 'confidence decisions are not the result of some inert post-decisional process, but reflect an online control process that moderates sensory evidence accumulation'. Why not use the fitted model to test if this 'online control process' would lead to the pattern shown in Figure 3d? My suggestion is to use the confidence model derived from the 'Replay' task to simulate the 'Stopping' task and stop the accumulation based on the online confidence estimate that the model produces. For example, for a target accuracy of 75% the decision would be stopped when the confidence is 75%. The authors could then test if the

simulations replicate the relationship between meta-d' and bound setting which they show in Figure 3d.

We thank the reviewer for this clever suggestion. We have conducted the simulation exercise described by the reviewer. This worked well considering the noise in parameter estimation, and the fact there was some shift in these parameters between experimental sessions. As observed for the human data, the relationship between the simulated bound efficiency and Type-II efficiency is $y = 1.22x + 0.33$ ($p = 0.001$; or excluding one outlier, $y = 1.6x + 1.7$, $p = 0.008$). Note that the Pearson and Spearman correlation coefficients are also significant, in reference to a later point raised by the reviewer (**R1C09**). The relationship between simulated bound efficiency and calculated bound efficiency was a little less obvious but still significant: $y = 0.78x + 0.08$, $p = 0.02$. Note that the Pearson correlation coefficient was significant ($p = 0.03$) whereas Spearman correlation coefficient was not significant ($p = 0.05$). We thank the reviewer again for prompting this validation of our model, which we now mention in the results of the revised manuscript commented **R1C05a, P10L302**:

“...furthermore, estimating the placement of Type-I bounds based on the Type-II evidence provided a good estimate of bound efficiency.”

And raising the implications of this in the Discussion, commented **R1C05b, P13L407**:

“Our interpretation goes further by postulating that the Type-II process is acting on the Type-I evidence accumulation online, as accumulation occurs. Indeed, we were able to reproduce observers' bound efficiency by simulating bounds based on the Type-II evidence estimate of the probability of correct responses (**Supplementary Materials, Figure S5**).”

The specific details have been added to the supplementary materials document (commented **R1C05, P06L131** in the **Supplementary Materials** document).

“We suggested that observers were using their confidence evidence to set and maintain the Type-I decision bounds. In this respect, the suboptimalities in confidence evidence are responsible for the suboptimalities in bound efficiency. By simulating optimal bounds on the confidence evidence (for example, stopping the Type-I accumulation when there was a 70% chance of a correct response based on the Type-II evidence), and then recalculating the bound efficiency based on the accumulated Type-I evidence, we were able to reproduce the relationship between bound efficiency and Type-II efficiency ($y = 1.22x + 0.33$ $p = 0.001$; or excluding one outlier, $y = 1.6x + 1.7$, $p = 0.008$; **Figure S5a**). The simulated bound efficiency was fairly well related to the bound efficiency estimated using observer's actual responses ($y = 0.78x + 0.08$, $p = 0.02$; **Figure S5b**).

Figure S5. Simulated bound efficiency. a) Bound efficiency was simulated based on parameters fit in the ‘Replay’ task, by using the Type-II evidence to estimate when to stop accumulating, and then calculating the corresponding bound on Type-I evidence. This provided the same relationship with observers’ Type-II efficiency as reported in **Figure 3d** of the manuscript. **b)** Relationship between simulated bound efficiency (based on parameters fit to the ‘Replay’ task) and bound efficiency calculated based on performance in the ‘Stopping’ task (completed in a different experimental session).”

R1C06 (6) Because there is no mixture of difficulty levels in the experiment, I would have expected that flat bounds would lead to a precision that does not change over time, as in the sequential probability ratio test. Why aren’t the optimal bounds flat? The supplementary materials say “A flat bound is only optimal under the assumption of constant costs in accumulating evidence, and constant evidence strength, which are not valid assumptions in our setting”. I do not see what makes these assumptions invalid. Is it because of the temporal weighting?

We thank the reviewer for pointing this out. The cited sentence was misleading and has been replaced in the revised manuscript. The optimal bound is indeed a flat (constant) bound on the likelihood of a correct response, which is not a flat bound on the *proportional* evidence. Indeed, as the observer accumulates more samples, they require less evidence *per sample* for a correct response, even without considering unequal temporal weighting of the samples. This has been clarified and commented **R1C06a**, **P03L61** of the **Supplementary Materials** document:

“However, in our experiment we sought to understand how human observers were setting their bounds relative to the optimal observer, in order to measure their efficiency in setting and maintaining Type-I decision bounds. The optimal observer sets a flat bound on the likelihood of a correct response, which is not necessarily a flat bound on the total accumulated evidence.”

We have also expanded the explanation in the Results, relating to reviewer #3’s comment 07 (commented with **R1C06**, **P05L157** in the revised manuscript):

“To maintain a constant probability of a correct response, the ideal bound decreases with the number of samples – i.e., a ‘collapsing’ bound (see Methods for more details). That is, as additional samples are accumulated, the observer requires less evidence per sample on average for the same probability of a correct response.”

We hope that the reviewer will find these clarifications useful.

R1C07 (7) The model has a parameter (α) that controls the weight of each sample of evidence, which explains why some subjects weight more the early samples (‘primacy’) and others weight more the late samples (‘recency’). In contrast, a more common way of explaining the patterns of ‘recency’ and ‘primacy’ is through covert bounds that end the accumulation before the end of the stimulus (eg, Kiani & Shadlen 2008), and/or leak in the accumulation that leads to recency effects (e.g., Tsetsos et. al. Front Neurosci 2012). In contrast, the current paper adopts a temporal-weighting function, which is not equivalent to a leaky integrator. For example, a temporary weighing function will affect the weight of each sample regardless of how many stimuli are shown, while the impact of the leak does depend on the number of stimuli. It would be helpful if the authors explain why they chose to model recency and primacy with a temporal weighting function instead of using leak and bounds. It would also be useful to see a histogram of the value of α across participants, to know how many of them show recency and primacy.

We understand the reviewer’s concern, and apologise for the imprecision of the description of primacy and recency effects in our model. The temporal weighting function we used actually corresponds to a leaky integrator. This was not explained well in the text and has now been modified in the revised manuscript, commented with **R1C07a, P18L555; R1C07b, P18L559; R1C07c, P18L561:**

“Functionally, during accumulation, the current accumulated evidence is weighted by α , before accumulating the next sample, so that when $\alpha > 1$ this creates a primacy effect and later evidence affects the decision less than the initial evidence. In contrast, when $\alpha < 1$ this creates a recency effect, and the observer’s decision places greater weight on the more recent evidence than the initial samples. Thus, by the end of the sequence, the weight on each sample n is equal to:

$$v_n = \alpha^{T-n} \quad (7)$$

Where T is the total samples in that trial and $n \in [1, T]$.”

Regarding the distribution of α across participants, 3 participants showed (slight) primacy in the stopping task, two of whom showed also (slight) primacy in the replay task. All other participants showed recency effects. The mean and standard deviation of the fitted parameters across observers are presented in Table S1 of the

Supplementary Materials document.

R1C08 (8) Figure S2. I don't understand how can the different curves cross each other in panel c (Participants 11 & 17). Shouldn't it be the case that the 'proportional evidence' is always higher for higher levels of confidence? (as seen for example for Participant 7).

The reviewer is correct that the observer should always require more evidence for a higher confidence rating. The fit in **Figure S2** (now **Figure S3** in the revised manuscript) was not based on an estimate of the observers' evidence, but on the ideal evidence (i.e., the difference in the probability of each category given the orientation). We had made this choice to understand – from a 'model-free' perspective – whether it would be appropriate to apply the same function for setting the boundaries used for confidence ratings. We have changed the y-axis label to better reflect this. We have now added a figure showing confidence as a function of the estimated evidence (i.e., based on the model fitted parameters), along with the fitted bounds, as suggested by reviewer 3 (**R3C20**).

Figure S3. Decision bounds and confidence criteria. **a)** The y-axis shows the proportional evidence (disrupted by samples of noise and leak – in this case, $\sigma = 1.5$, $\alpha = 0.9$) which is the accumulated evidence divided by the number of samples the evidence was accumulated over. The dashed lines show the average proportional evidence where the optimal observer chose to respond whilst the solid lines show the fit of an exponentially decreasing function. **b)** The same simulated data as in **a)** is shown with the fit of a power function. The simulations show a steeper decrease than can be captured by the power function (the simulated optimal observer is more cautious with fewer samples and more liberal with more samples). **c)** The y-axis shows the proportional ideal accumulated evidence (not disrupted by noise and leak). The dashed lines show the proportional evidence by the number of samples for each confidence rating (three observers in separate plots). The solid lines show the fit of the exponential function. **d)** Model fitted bounds on the observer's estimated proportional evidence based on parameters fitted to the

confidence decisions. Dotted lines show the average proportional evidence for each confidence rating.

R1C09 (9) In Figure 3d, why did the authors used the “line that minimizes the perpendicular distance” instead of the correlation coefficient? Is the correlation coefficient significantly positive for the data in Figure 3d?

We made this choice because we considered the line that minimises the perpendicular distance to be both non-parametric and meaningful. Indeed, a slope of 1.5 literally means that we expect an increase of 0.5 in bound efficiency with an increase of 0.75 in Type-II efficiency. The metrics are comparable: ideal Type-II efficiency is 1, and ideal bound efficiency is 1, so the scale is important as well as the relative position. In comparison, the Spearman correlation coefficient tells us how similar the ranks of bound efficiency are to the ranks of Type-II efficiency – which effectively ignores the scale by considering only the ranks on each dimension. Nevertheless, we can reassure the reviewer that the test statistic for the correlation between the two measures is significantly positive no matter the method used: Pearson (all): $r = 0.4$, $p = 0.041$; Pearson (excl. outliers): $r = 0.45$, $p = 0.029$; Spearman (all): $\rho = 0.39$, $p = 0.047$; Spearman (excl. outliers): $\rho = 0.40$, $p = 0.049$.

R1C10 (10) Typo on page 11 line 26: “process in terminated” -> “process is terminated”

This typo has been fixed, thank you. Commented **R1C10**, **P11L344**:

“...the sensory evidence accumulation process is terminated by a stricter bound...”.

Reviewer #2:

In this manuscript Baldson and colleagues are presenting the results of a study in which they analyse the role of confidence in perceptual evidence accumulation. I really liked this study, the task manipulation is very clever and the analyses are thorough (all statistical controls and out of sample validations were performed correctly). The manuscript and figures are very clear and easy to understand.

R2C01 The model they developed is simple but insightful and it is used with great care to dissect the subtleties of their behavioural results. The authors collected also pupillometry data to complement the behavioural data. These pupillometry results are in my opinion the least interesting set of data presented here but by no mean of no interest. Maybe the authors might

want to discuss more (even at speculative level) on the possible role of noradrenaline (measured indirectly via pupil dilation) in modulating the circuit they proposed in fig 5

We agree with the reviewer that this is an interesting question, and the possible role of noradrenaline (measured indirectly via pupil dilation) should be investigated further in future research. We think that the effect of confidence on pupil dilation/constriction may reflect different mechanisms, and more work would be needed to arbitrate between them. For example, the faster pupil constriction observed after the response may be triggered by faster task disengagement, where observers disengage from the task sooner when they are more confident. A note to this effect has been added to the discussion, commented **R2C01, P13L391**:

“Whilst phasic pupil dilation was associated with crossing the decision bound (temporally dissociated from the time of the response in the ‘Replay’ task), a distinct effect was seen related to confidence: faster pupil constriction following the response (possibly due to an indirect effect of confidence on task disengagement).”

Our use of pupil dilation data was solely aimed at showing that the transient peak of pupil dilation found in the ‘Stopping’ task at decision time was shifted earlier in time in the ‘More’ condition when subjects had crossed their covert decision bound seconds before response execution.

The most interesting finding is that confidence provides an online control on the accumulation of perceptual evidence that triggers first order choice response. The idea that confidence is not simply a readout of choice is not new but, as far I can tell, this work provides the first compelling evidence that this ‘metacognitive’ control works from the very incipit of the evidence accumulation process and exerts an online control on type I accumulation. Given the high variability in metacognitive abilities between individuals it would be interesting to see if between subject variability in meta-d’ accounts for inter-individual differences in this online control process. However since the study includes only 20 participants it might be hard to detect these between subject effects. However this is not a limitation of the current study and I am sure that idea can be picked up in future studies. I believe that this is an important study that will spark interest in the community and I am therefore very happy to recommend it for publication in Nature Communications.

R2C02 One more thing: I could not find a link to the data or to the analyses scripts. I know that some of the authors share my enthusiasm for transparent and open access science and I would like to kindly nudge them to make their analyses and data available on a sharing platform like GitHub. This would certainly increase the impact of their work.

It is unfortunate that the reviewers were not given access to the data and code that we provided to Nature Communications during submission of the original manuscript. We agree with the reviewer, and have planned to upload the data and code to Open Science Framework on a fork of the pre-registration once the manuscript is accepted for publication.

Reviewer #3:

This is a strong and interesting manuscript in which the authors weave together well-designed experiments, computational modeling, and pupillometry to make substantive contributions to the field's growing understanding of the relationship between perceptual (type-I) and metacognitive (type-II) processing. Most provocatively, they demonstrate a correlation between efficiency in perceptual decision making and metacognitive efficiency that adds to a growing body of evidence suggesting how metacognition not only evaluates perceptual processing, but may also directly participate in and shape it (Fig 3d). They additionally provide modeling and pupillometry evidence that subjects set covert decision bounds, neglecting to consider additional stimulus evidence once a decision has been made (Fig 4). Rounding things out, they demonstrate a very clean dissociation between pupillometric signatures of perceptual decision-making and confidence (Fig 6) and perform additional modeling analyses to flesh out a bigger-picture view of how type-I and type-II processes may be related (Fig 5).

R3C01 To my mind, the most interesting aspect of this manuscript is the demonstration of a link between optimality in metacognition and optimality in perceptual decision-making, which could be interpreted as potentially being a tell-tale sign of a type-II process exerting top-down control on a type-I process. However, one consideration that significantly moderates my enthusiasm is that the 'Stopping' task requires subjects to provide a type-I response once they've reached a certain threshold of estimated probability that their response will be correct, given the evidence accumulated thus far. But this means that the very nature of the task requires subjects to yoke the type-I response ("I think the stimuli were drawn from the blue distribution") to a type-II judgment ("I will only provide my type-I response once I think I'm 80% likely to be correct"). So, perhaps it is not so surprising that the subjects who are poor at distinguishing correct from incorrect responses with a 4-point confidence rating scale (poor type-II efficiency) are also poor at producing an absolute probability estimate of how likely they are to be correct (poor bound efficiency). More broadly, I worry that the type-I / type-II correlation could possibly be a contrived consequence of the task design and instructions in the 'Stopping' task, rather than reflecting a more general relationship that can be observed in more naturalistic or open-ended task settings

that do not so explicitly encourage subjects to yoke type-I responses to type-II judgments in this way. I do not think this is a fatal objection or a completely trivializing observation, but it should be addressed somewhere in the Discussion.

We thank the reviewer for outlining this concern. Indeed, it has been difficult to sufficiently emphasize that our results are by no means trivial, although some readers may initially find the relationship unsurprising. In our manuscript we provide the first test of the relationship between the observers' ability to efficiently accumulate perceptual evidence (bound efficiency) and their metacognitive insight into their own perceptual performance (Type-II efficiency). We show this by explicitly setting target performance levels and measuring how well observers adjust their bounds in the face of these performance targets. If there were no relationship between Type-II and bound efficiency in this task, then one would not expect to find the relationship in tasks where the need to estimate accuracy for the purpose of implementing bounds is less explicit. We interpret that the reviewer's main concern is that these results may not generalise to more naturalistic tasks where the observer is not explicitly asked to set their bounds according to a performance target. It is true that we cannot demonstrate this with our data, and we have added a comment to the discussion to emphasise that this should be examined in future research (**R3C01a, P13L414**, see quoted text below).

There are two ways a reader could interpret the relationship between Type-II and bound efficiency as trivial. First, it could be that some observers are better across all experimental measures (because of motivational factors, or their ability to concentrate throughout). We address this first issue with two comparisons: (i) the relationship between observer's Type-I sensitivity in the 'Replay' task and bound efficiency in the 'Stopping' task, and (ii) the relationship between Type-II efficiency in the 'Replay' task and the magnitude of the suboptimalities in evidence accumulation in the 'Stopping' task. There was no relationship between these variables, so it is not the case that there is a general correlation between different performance measures across tasks. This we emphasise in the discussion, commented **R3C01b, P13L397**, quoted below.

Second, it could be that observers are only using their Type-II evidence to set and maintain their bounds efficiently because we instructed them to (and otherwise there would be no relationship). This second issue is conceptual. At the surface, one could suggest that it is unsurprising that observers who are poor at discriminating correct from incorrect responses with confidence ratings should also be poor at estimating the likelihood that they are correct, and therefore show poor bound efficiency. But the broader theoretical context suggests the relationship is not so simple. The current normative description of confidence suggests (at least implicitly) that it is not computed online: that only after the decision is made does confidence take the perceptual evidence and convert it into an estimate of the likelihood of being correct. In order to use confidence evidence to implement bounds, confidence must be computed online. If confidence were computed only after the decision, we may not have found a significant relationship between Type-II efficiency and bound efficiency. The current theories of how observers implement bounds suggests the involvement of learning mechanisms to balance speed and accuracy. The focus of the discussion has been on how observers are able to flexibly adjust their bounds in the face of different costs or speed pressures, whilst the importance of estimating accuracy has been minimised. If bounds were set based on an estimate of accuracy from some separate learning mechanism, we may not have found a significant relationship between Type-II efficiency and bound efficiency. Further, our analysis describes observers'

ability to efficiently adjust their bounds: bound efficiency is calculated as the change in bound between 70% and 85% target performance conditions, divided by the ideal change in the bound. This measure emphasizes perceptual precision over perceptual accuracy. Observers can over- or under-estimate their accuracy but still have good bound efficiency so long as they optimally adapt their bound. This measure focusses on the implementation of the bound, minimising the importance of actually meeting the performance targets. We have explained this more fully in the results, commented **R3C01c, P06L196**:

“Bound efficiency’ was then calculated as the change in bound across conditions that the observer actually implemented, divided by the change in bound they should have implemented if they were to reach the target performance levels. This expected behaviour was obtained by simulations given the other suboptimalities affecting observers’ performance (see **Methods** for more details). In this way, an observer with high bound-efficiency could over- or under-estimate their performance (poor accuracy relative to the target performance levels), though they still *adjust* their bounds appropriately in the face of different performance targets (good precision).”

Thus, the strong relationship between bound efficiency and Type-II efficiency suggests that the ability to flexibly adjust bounds strongly relies on the representation of the likelihood of being correct that is generated by the same metacognitive system as post-decisional confidence judgements. We measure Type-II efficiency and bound efficiency in separate tasks, in separate sessions. We show that Type-II decisions are based on a partially dissociable accumulation process to Type-I decisions, with dissociable effects on pupil size. Yet these distinct measures, based on distinct processing, measured in separate sessions, are strongly correlated.

Here is the relevant section of the discussion:

“Despite all these differences between Type-I and Type-II evidence accumulation, there was a surprisingly strong correlation between observers’ Type-II efficiency and their ability to efficiently set and maintain Type-I decision bounds. It is unlikely that this correlation emerges indirectly due to some common underlying variable, such as the observers’ motivation in the experiment in general, because of the lack of a relationship between observer’s Type-I sensitivity (which should also vary with task commitment) in the ‘Replay’ task and bound efficiency in the ‘Stopping’ task. Likewise, there was no relationship between Type-II efficiency in the ‘Replay’ task and the magnitude of the suboptimalities in evidence accumulation in the ‘Stopping’ task. The strongest interpretation of the correlation is that there is a causal relationship between Type-II efficiency and bound efficiency: Type-II evidence is being used to set and maintain boundaries on Type-I evidence accumulation (though a test for causality is left for future research). This ‘metacognitive control’ has been previously suggested to operate in a post-decisional manner: the current confidence will influence future Type-I evidence accumulation^{37,38,39}. Our interpretation goes further by postulating that the Type-II process is acting on the Type-I evidence accumulation online, as accumulation occurs. Indeed, we were able to reproduce observers’ bound efficiency by simulating bounds enacted based on the Type-II evidence estimate of the probability of correct responses (**Supplementary Materials, Figure S5**). This kind of interaction would readily explain how observers actively seek more information when they are uncertain⁴⁰, and how observers can integrate loss functions into their Type-I decisions⁴¹. In addition, this kind of relationship offers a mechanism by which decision bounds could be implemented at the neural level, which is supported by evidence showing activity in the dlPFC is modulated

according to changes in the speed-accuracy trade-off^{42,43,44} and according to metacognitive confidence^{45,46}. However, it remains to be tested whether observers utilise their Type-II evidence for bound-setting in other contexts.”

We hope that the reviewer will find these additions useful.

R3C02 Although I do not count this against the current manuscript, some gaps in the logic of the overall argument that could be filled out in future research are: (1) the computational modeling does not incorporate a mechanism that reflects a possible direct interaction between type-II efficiency and the setting of type-I decision bounds; (2) ideally, there would be a way to test for a causal relationship between type-II processing and type-I decision making. It may be worth discussing these points briefly in the Discussion as areas for future work to address.

These are indeed essential points for future research, which are all now present in the discussion of the revised manuscript. Point 1 is discussed at **P13L411**, commented **R3C02a**. The simulations suggested by reviewer #1 (**R1C05**) suggests that the bound could be set directly on the estimates based on the Type-II evidence, though this does not directly clarify the neural mechanisms. Point 2 has now been noted in the discussion, commented **R3C02b**, **P13L404**, incorporated into the quote above.

Overall I think this is a strong, well-composed, thought-provoking work that provides a valuable stepping stone in the literature for future studies to more closely investigate the important and fascinating topic of possible direct and online interactions between type-I and type-II processes.

Below I provide a more detailed set of smaller points pertaining to details of the analyses, arguments, and presentation.

Results

R3C03 It would be helpful to mention in the Results that the statistics are for n=20 subjects. This is a basic and important enough detail that one shouldn't have to refer to the Methods to find it.

We agree with the reviewer. The number of tested participants has been added, commented as **R3C03**, **P3L77** in the revised manuscript:

“Across all three Type-II contexts, observers (N = 20) made the same Type-I decision: whether...”

R3C04 The authors report confidence intervals for proportion correct, but seem to perform all statistical tests on d' , which caused me some initial confusion. Given that the statistical analyses were performed on d' , it would seem more pertinent to have confidence intervals reported for d' instead of, or in addition to, proportion correct. It would also be relevant to know if observers exhibited any response bias, which could influence the proportion correct data.

We chose to describe the task to participants in terms of proportion correct because this is more comprehensible. We actually chose the target performance levels to be $d' = 1, 2, \text{ and } 3$, and these translate to approximately 70%, 85%, and 90% correct. We describe first the proportion correct, and use this for graphical purposes to match the participant instructions, and avoid the reader having to do conversions. Statistical analyses were performed on d' because Type-I sensitivity was our actual variable of interest. Confidence intervals on d' have been added, commented as **R3C04a, P04L114** in revised manuscript:

“This corresponded to a Type-I sensitivity (d') of 1.2 [± 0.17], 1.7 [± 0.14], and 1.9 [± 0.19] in each target performance condition”.

A note on the criterion has been added to the methods, commented **R3C04b P19L595**:

“There was also little evidence for response bias from the behavioural data (mean criterion = 0.03 [± 0.05]). ”

R3C05 It is unclear why the authors choose to use the non-parametric Wilcoxon sign rank test when comparing d' across conditions, rather than a parametric t-test which would likely offer greater statistical power. If the concern is about how the assumptions of the t-test might be inaccurate for the sampling distribution of d' , the authors could refer to the following work to more accurately perform statistical inference on d' : Miller, J.O. (1996). The sampling distribution of d' . *Perception & Psychophysics*, 58, 65–72.

We thank the reviewer for pointing out this reference. This is an intricate question, and we agree with the reviewer that it is important to motivate the use of statistical tests. Our choice of non-parametric tests not only concerns the assumptions of parametric tests, but the theoretical nature of the test itself. Within-subject parametric tests ask whether two dependent samples were drawn from the same distribution, whilst the non-parametric Wilcoxon signed rank test asks whether the difference in dependent samples is symmetric around 0. In many cases, the null hypothesis of t-tests is more restrictive than the true null – we don't care if it's the same distribution, we care if the difference is 0. There is also the problem of the assumptions of the t-test, which is calculated based on the mean and variance of the t-distribution. With small sample sizes, the estimate of the

mean and variance of the population is poor. Even so, small sample sizes tend to be problematic for t-tests even when it is 'known' that the population distribution is normal. The assumption of normality tends not to be a problem for large sample sizes because of central limit theorem, which states that independent random samples drawn from any shaped (but identically distributed) distributions tend towards normality. But central limit theorem relies on the assumption that samples are drawn from identical distributions. Are samples of d' identically distributed? No, equation 10 of Miller (1996; equation 6 of Gourevitch and Galanter, 1967) shows that the variance of d' is dependent on d' (also visible in Table 2 of Miller 1996). If d' is calculated over a large enough number of trials, the difference in variance across d' is small, and central limit theorem may hold. But here we only had 100 trials to calculate d' . There is no guarantee that the group is normally distributed, and with a sample size of 20, the test for normality is unreliable. We therefore prefer non-parametric tests over group-level d' when the sample is small, for both practical and theoretical reasons.

R3C06 Perhaps related to the above, I found it odd that the reported confidence intervals for proportion correct in the 85% and 90% conditions of the Stopping task (0.80 ± 0.021 and 0.82 ± 0.023) were nearly identical to those of the 'Free' and 'More' tasks (0.80 ± 0.025 and 0.82 ± 0.025), and yet the results of the Wilcoxon signed rank tests yielded a significant difference in the former case ($Z=2.35$, $p=0.037$) but was nowhere close to significant in the latter ($Z=0.04$, $p=0.68$). I realize the statistical tests were performed on d' rather than proportion correct, and that the 'Free' and 'More' data were slightly noisier, but the stark difference in the Wilcoxon tests is still somewhat surprising. Do these two statistical comparisons look more similar if they are assessed with a parametric test rather than non-parametric? This is not an entirely trivial point since the authors interpret the lack of statistically significant difference in task performance between the 'Free' and 'More' conditions as one piece of evidence for the claim that subjects set covert decision bounds that terminate evidence accumulation prior to perceiving all the additional stimuli in the 'More' condition. Even if an alternative test did reveal a significant difference between the 'Free' and 'More' conditions, I don't think it would make all that much of a difference to the authors' overall argument, but it would nonetheless be informative to get clearer on all this.

This is a meaningful concern, also raised by reviewer #1 (**R1C03**), which is due to a lack of sufficient description in the original manuscript. Indeed, we left an important detail to the methods section (**P15L462**, and **P15L488**), which can explain the apparent discrepancy pointed out by the reviewers. We have tried to explain this detail better in the results now, whilst maintaining the brevity required by *Nature Communications* (commented **R3C06a**, **P03L81**; **R3C06b**, **P04L129**; **R3C06c**, **P04L134**). This detail is that the same 100 pre-defined trials for each participant were repeated throughout the entire experiment. In the 'Free' task, observers completed each trial three times. Therefore, for each trial, there were three responses (3 repeats) and three 'number of samples' ('Less', 'Same', and 'More' conditions). For the 'Less' condition of the 'Replay' task, we took the *minimum* number

of samples minus 2 samples. For the 'Same' condition, we took the *median* number of samples. And for the 'More' condition, we took the *maximum* number of samples plus 4 samples. For clarity, an explicit example of this procedure has been added to the methods, commented **R3C06d** on **P15L490**:

"For example, if the observer chose to respond after 4, 5, and 10 samples on the three repeats of one pre-defined trial, they would be presented with 2, 5, and 14 samples in the 'Less', 'Same', and 'More' conditions of the 'Replay' task respectively."

The comparison of d' was not conducted across all trials of the 'Free' task (300), but for the 100 trials chosen for each condition ('Less', 'Same', and 'More') in the 'Replay' task. Thus, the comparison asks if precisely 4 additional samples improve d' , not if sensitivity is improved in the 'More' condition relative to the 'Free' task taken as a whole. This has been clarified and commented as **R3C06e**, **P05L135**:

"In the 'Less' condition, two fewer samples corresponded to a substantial decrease in performance within subjects ($M = 0.69 [\pm 0.025]$, Z ('Less' vs. 'Free' d') = 3.51, $p_{\text{bonf}^*3} < 0.001$) but in the 'More' condition, four additional samples did not significantly improve performance within subjects ($M = 0.82 [\pm 0.025]$; $Z = 0.04$, $p = 0.68$)."

Importantly, when we compare d' in the 'More' condition to d' calculated across all trials of the 'Free' task, the difference is also not significant ($Z = 1.34$, $p = 0.18$). Note, also, that because our test is a within-subject comparison, it is important to consider the standard error of the difference. In the 'Stopping' task, d' was on average 1.74 for an 85% target and 1.92 for a 90% target, corresponding to an average difference of 0.18 and a 95% within-subject confidence of 0.064 (yielding a significant difference). In the 'More' condition, d' was on average 1.85 and in the 'Free' task (all trials) it was on average 1.75, corresponding to an average difference of 0.095 (about half the above) and a 95% within-subject confidence of 0.068. In other words, there is a substantially smaller within-subject effect size for the difference between the 'More' condition and the 'Free' task than between the 90% and the 85% target conditions within the 'Free' task.

Using a parametric t-test, neither comparison is significant (all trials: $t(19) = 1.42$, $p = 0.17$; matched +4 trials: $t(19) = 0.78$, $p = 0.44$). We hope this alleviates the reviewer's concerns.

R3C07 p. 5: "In order to achieve the performance targets, the observer imposes a bound on the accumulated evidence, which decreases with the number of stimuli— i.e., a 'collapsing' bound (see Methods for more details)."

While reading the Results, I had in the back of my mind the question of why collapsing rather than fixed bounds were used. The justification is sensible-- simulations show that ideal observer must

use these bounds in order to yield the target task performance. It would be helpful to briefly state this justification in the Results, referring the reader to the Methods for more details, rather than burying the first mention of this point much deeper in the manuscript.

We thank the reviewer for this suggestion, which should also help to address the comment of reviewer #1 (**R1C06**). This justification has been added to the results, commented **R3C07, P5L157** in the revised manuscript:

“To maintain a constant probability of a correct response, the ideal bound decreases with the number of samples— i.e., a ‘collapsing’ bound (see **Methods** for more details). That is, as additional samples are accumulated, the observer requires less evidence per sample on average for the same probability of a correct response.”

R3C08 The result plotted in Fig 3d as described in the figure legend is different from that reported in the main text of the Results, apparently due to omitting two subjects from analysis in the former but not the latter. Why the discrepancy? It is fine to only show the single result in Fig 3d, but both results should be described in the Results, with an expanded justification for omitting two subjects in the modified analysis-- the justification provided in the figure legend is not entirely clear.

Additionally, presumably the p-value reported in the figure legend pertains to the statistical significance of the slope of the fitted equation, is that right? Please clarify in the legend.

This is correct: the p-value pertains to the significance of the slope of the fitted equation. The discrepancy is due to whether the outliers are included or not. As requested, the result has been added to the main text (commented as **R3C08a, P07L207** in revised manuscript) and the statement in the legend has been clarified (**R3C08b, P07L226**):

“Each observer’s bound efficiency (x-axis) by their Type-II efficiency (y-axis) and the line minimising the perpendicular distance from these points, here shown excluding two outlier participants (open circles) whose optimal bound was estimated using maximum performance rather than 85% correct ($y = 1.5x + 0.2$, $p(\text{slope}>0) = 0.026$, with outlier participants removed).”

R3C09 p. 7: "We found the suboptimalities in evidence accumulation were insufficient to explain the lack of increase in performance in the ‘More’ condition relative to the ‘Same’ condition (see Supplementary Materials for details)."

I find the logic of the argument referred to in the Supplementary Materials here puzzling. It seems to run as follows: (1) when we fit parameters separately for the 'Same' and 'More' conditions, the

'More' condition had more noise and less leak; (2) it is unrealistic for these parameter values to be different, given that the 'Same' and 'More' conditions were randomly interleaved and subjects could not know ahead of time which trial was from which condition; (3) therefore differences in noise and leak cannot explain why task performance was not much better in 'More' than in 'Same'.

I think (2) is a valid point, but pointing out (2) means performing step (1) is superfluous. Furthermore, the conclusion (3) does not seem to address the main point, which is the possibility that existing levels of noise and leak in the 'Same' condition might fundamentally constrain how much 4 extra samples in the 'More' condition can improve task performance.

I think a far more informative analysis would be as follows. First, state point (2), that it is a priori unreasonable to fit different noise and leak parameters for the 'Same' and 'More' conditions. Use this as justification for fitting noise and leak only to the 'Same' condition, in both the covert-bound and no-bound versions of the model. Using the parameters from this fit, perform simulations to see how much each model predicts task performance should improve, on average, by presenting 4 additional evidence samples, and assess how this compares to the actual data. If the no-bound model predicts that task performance should improve substantially more than it actually does in the data, then you are justified in concluding that "the suboptimalities in evidence accumulation (i.e. the noise and leak fitted to the 'Same' condition) are insufficient to explain the lack of increase in performance in the 'More' condition relative to the same condition." However, if the no-bound model predicts that the increase in task performance is numerically similar to what is actually observed, then it would seem that the suboptimalities in evidence accumulation (i.e. the noise and leak fitted to the 'Same' condition) ARE indeed sufficient to explain the lack of task performance increase.

Of course, either way, the model comparison analysis in Fig 4b is still good evidence that even if the no-bound model is *plausible* to explain the data, the covert-bound model is still *better*.

We thank the reviewer for so carefully outlining his reasoning, which fully aligns with ours. When simulating responses in the 'More' condition based on the fit to the 'Same' condition, performance still did not significantly increase, as we previously had assumed it would. We have therefore replaced the comparison of model parameters (commented **R3C09a, P08L234**) with the model comparison test: the employment of a covert bound is a better explanation of the behavioural performance than the suboptimalities affecting the 'Same' and 'Less' conditions alone. As the reviewer suggests, we have made this argument more apparent in the Results (commented **R3C09b, P08L238**). We have also removed the section from the supplementary comparing the parameters fit across separate conditions (commented **R3C09** in the **Supplementary Materials**, removed at

P09L181). As well as the comment on this analysis in the discussion (commented **R3C09c**, **P12L355**). Although the pattern of changes in the parameters is also suggestive of a covert bound, the reviewer is correct in pointing out that the argument does not logically follow that the parameters fit to the ‘Same’ condition are insufficient. The relevant section of the manuscript now reads:

“In the ‘Replay’ task, the additional samples in the ‘More’ condition should have driven increased performance compared to the ‘Same’ condition, but from the statistical analysis reported in preliminary analysis, this was not the case (**Figure 4a**). This lack of improvement has two possible explanations. First, there could be a performance limit, due to the suboptimalities in evidence accumulation parameterised by σ (inference noise) and α (inference leak). Second, and more interestingly, observers could be employing a *covert bound* on evidence accumulation, where observers do not accumulate evidence beyond this bound, even when additional evidence is available. To compare these hypotheses, we fit two models, one with and one without a covert bound, fixing the suboptimalities to those fit to the ‘Same’ and ‘Less’ condition. There was a significant improvement in model fit with a covert bound relative to the model without a covert bound, assessed using a five-fold cross validation (mean relative increase = 0.047, $Z = 2.46$, $p_{\text{bonf}^*3} = 0.041$; as shown in the left-most bar of **Figure 4b**), suggesting that the employment of a covert bound is a better explanation of the behavioural responses than the suboptimalities in evidence accumulation alone. This model comparison included two other models in which a covert bound was fit to Type-II evidence accumulation (more details below) of which neither model showed significant evidence of an improvement in fit (for a ‘hard’ bound, the same as Type-I; $Z = 1.79$, $p = 0.073$; nor an independent bound $Z = 0.11$, $p = 0.91$). Further details on reflexive vs. absorbing²⁹ covert bound comparisons can be found in the **Supplementary Materials (Figure S7)**.”

R3C10 Fig 4b, please label the y-axis more clearly. Presumably what is being quantified here is something like change in log likelihood?

This has been changed to ‘Difference in log-likelihood’, commented **R3C10, P09L267**:

Figure 4. Evidence for a covert bound on Type-I evidence accumulation. a) Average proportion correct in each condition of the ‘Replay’ task. In the ‘Same’ condition, observers were shown the same number of samples as they had chosen to respond to in the ‘Free’ task, in the ‘Less’ condition they were shown two fewer samples, and in the ‘More’ condition they were shown four additional samples. The red horizontal line shows the average proportion correct in the ‘Free’ task, where the exact same trials were shown (with the exception of the condition manipulation). Error bars show 95% within-subject (thick) and between-subject (thin) confidence intervals. Open red circles show the model predictions based on simulations of the bounded model. **b)** Difference in log-likelihood (relative improvement in fit) between the model fit with a covert boundary compared to the model without. The bars show the improvement in the fit to Type-I responses with a bound on Type-I evidence accumulation (labelled Type-I), the improvement in the fit to Type-II responses when the Type-II evidence stops accumulating at the same time as the Type-I evidence (labelled Type-II), and the improvement in the fit to Type-II responses when an independent bound is fit to the accumulation of Type-II evidence (labelled Type-II ind.). Error bars show 95% between-subject confidence intervals. **c)** Average standardised pupil size in the ‘Stopping’ task (top) and ‘Replay’ task (bottom), separated by conditions, and averaged across time-windows aligned to the start of each trial (left) and to the response (right). Shading shows 95% between-subjects confidence intervals. Trial-start aligned data were baselined to the average at time 0, response aligned data were baselined to the average prior to trial start (-4 to -3 s in the ‘Stopping’ task, and -6 to -5 s in the ‘Replay’ task). Black vertical lines in the trial-start aligned plots show the timing of the 2nd and 3rd samples. Vertical lines in the response-aligned plots show the average median timing of the start of the trial in each condition. Differences from baseline in the response-aligned epochs were tested using Wilcoxon sign rank tests, with significant clusters shown with horizontal lines.

R3C11 In Fig 5, leak is illustrated as alpha but described in the legend as nu. To this point in the manuscript, I believe nu has only been mentioned briefly in the legend of Fig 2, and the relationship between nu and alpha is not discussed at length til the Methods. The discrepancy between the presentation here using alpha in the figure and nu in the legend, is potentially confusing if one has not yet read the methods. It would be helpful to briefly clarify the relationship between alpha and nu in the figure legend.

We thank the reviewer for catching this error. The legend was not properly updated from a previous version of the figure, and now refers to alpha, commented as **R3C11, P10L309** in revised manuscript:

“Figure 5. The accumulation of evidence for Type-I and Type-II decisions. The dashed box encloses the hidden decision processes, which determine the relationship between the observable variables (the physical input and behavioural output). Information from the physical stimulus is transformed into decision evidence, which is accumulated for the Type-I decision with additive Gaussian noise (ϵ_s) and weighted according to the leak (α_s). Evidence accumulated for the Type-II decision incurs additional noise (ϵ_c) and a separate leak (α_c). Type-II control

is exerted on the Type-I evidence accumulation process, depicted by the red arrow, where the accumulated evidence is sent to the decision output once the boundary is reached, based on the Type-II evidence. The Type-II evidence may continue to accumulate even after the boundary has been reached.”

R3C12 p. 9: "...we found both σ and α to correlate when allowing model parameters to vary independently across the Type-I and Type-II evidence accumulation processes..."

This is the only mention in the manuscript of a modeling approach that directly reflects the model structure of Fig 5. However, the given explanation of the modeling approach is brief and not entirely clear, and it is not described more fully in the Methods. Presumably what the authors mean is that they modeled two independent evidence accumulation processes, call them z_I and z_{II} , such that type-I response was determined by z_I crossing the covert bound, whereas confidence was determined by where z_{II} fell relative to the three confidence bounds. Is that right? It is also not clear when the readout of confidence from z_{II} occurred in this modeling approach. Did it occur at the same time as z_I crossed the decision bound, or did it occur after the last presentation of the sample? It seems appropriate to give a fuller characterization of this modeling approach in the Methods.

We thank the reviewer for highlighting this point that was not completely clear. In the terminology of the reviewer, we fitted models where z_I and z_{II} were the same and different. For models where z_I and z_{II} were different, we either fit an entirely independent set of parameters, or used a ‘partially correlated’ design, where Type-I parameters were fixed, but the Type-II could incur additional noise and/or a separate leak (and all combinations). Model comparison suggested the most parsimonious explanation of the data was one of these partially correlated models (**Figure 5**). We tested whether there was a bound on Type-II accumulation (**Figure 4B**) and found no evidence for one, so z_{II} incorporated evidence from all samples. These points have now been clarified, commented as **R3C12, P21L636**:

“First, a model was fit using the same z for Type-I and Type-II responses. Entirely separate parameters (leading to independent z for Type-I and Type-II responses) were fit in the ‘parallel’ model, where the Type-I parameters were fixed to those fit to only the Type-I responses. In partially correlated models some parameters for the Type-II z were fixed to be the same as those affecting the Type-I z . These models compared all combinations of fixed/variable noise and leak, and compared whether additional noise was added with each sample of evidence, or a single sample of noise irrespective of sequence length. Model comparison showed a partially correlated model, where Type-II z is affected by additional noise and a different leak, best accounted for Type-I and Type-II responses. We then compared models in which the observer accumulates Type-II evidence over all samples and models implementing a bound on Type-II evidence accumulation (either the same bound as the Type-I bound, or an independent bound). There was no evidence for a Type-II accumulation bound (**Figure 4b**); Type-II z

accumulated evidence across all presented samples.”

R3C13 It may be worth noting that this model in which type-II decisions inherit noise from type-I processes, while receiving additional type-II noise is consistent with other models put forth in the literature, e.g. Mueller & Weidemann, 2008; Jang, Wallsten, & Huber, 2012; De Martino, Fleming, Garrett, & Dolan, 2013; Maniscalco & Lau, 2016; Rahnev, Nee, Riddle, Larson, & D’Esposito, 2016; Berg & Ma, 2016; Bang, Shekhar, & Rahnev 2018.

We have edited the sentence on **P12L369** to better reflect this, commented **R3C13**:

“This is consistent with other findings in the literature suggesting that observers Type-II responses inherit noise from the Type-I process³², and may continue to accumulate Type-II evidence even after their Type-I response with additional Type-II noise^{8,16},”

R3C14 p. 10: "... (Figure 6c, from -0.2 s relative to the response, $p < 0.002$)"

This is referring to Fig 6b, not 6c.

Thank you, this has been fixed, commented as **R3C14**, **P10L318** in revised manuscript

R3C15 It would be interesting to briefly note somewhere in the manuscript, perhaps in the Supplementary Materials, what values of meta-d' and meta-d'/d' were produced by the best/most comprehensive model when fitted to the Replay data. Were they at all close to the empirical values?

Simulated meta-d' was strongly correlated with participant meta-d' based on 1000 simulations of the model for each observer ($y = 0.74x + 0.34$, $p < 0.001$; or Spearman's $\rho = 0.59$ $p = 0.007$; or Pearson's $r = 0.71$, $p < 0.001$). A note on this was added to the results, commented **R3C15**, **P10L300**:

“By simulating confidence ratings using the fitted parameters, we were able to closely estimate observers' Type-II sensitivity ($\rho = 0.59$ $p = 0.007$)”

Methods

R3C16 There is some ambiguity in how the variable T is defined that should be cleared up. On p. 17, following eq. 1, T is identified with the number of samples for a given trial, such that T varies from trial to trial. This makes it sound as if T is the total number of samples that end up being displayed on a given trial.

However, when T appears in the equations (eqs. 2, 5, 7, 8), it seems to denote not "the eventual number of samples shown during this trial," but rather "the number of samples that have been shown *up to and including the current sample.*" This would mean that T varies not only *between* trials, but also across time *within* a trial. This is especially important for understanding eq. 7, since a value of T that changes over time to indicate the total number of samples seen so far would be needed to capture the intended "leak" effect in ν_n . For instance, at the first sample of a trial, $T=1$ and the weight placed on the first sample $\nu_1 = \alpha^0 = 1$. By the tenth sample of a trial, $T=10$ and now $\nu_1 = \alpha^9$, a much lower value than 1 when $\alpha < 1$.

We apologise for not making this clearer. T always refers to the total number of samples in a trial, and n to the current stimulus. Two statements have been added to make this clearer, commented as **R3C16a**, **P17L543** and **R3C16b**, **P18L561**. In equation 7, T is the final number of samples. With each additional sample, the evidence is multiplied by α . If $\alpha = 0.5$, and the first sample is decision evidence of 1, by the time the next sample is accumulated, the first sample is now only 0.5, by the time a third sample is accumulated the first sample will contribute 0.25, and after a fourth sample, 0.125, such that if only four samples were accumulated ($T = 4$) the first sample ($n = 1$) would finally be worth $1 \times 0.5^3 = 0.125$ ($1 \cdot \alpha^{(T-n)}$).

R3C17 p 18: "Functionally, the observer weights the current accumulated evidence by alpha, so that..."

This seems incorrectly stated, as according to eq 8, evidence sample n is weighted by $\nu_n = \alpha^{(T-n)}$, not alpha. Furthermore, if T denotes the current sample, then the weight placed on the current sample (which is presumably what "current accumulated evidence" in the quoted sentence is referring to) would always be $\nu_T = \alpha^{(T-T)} = 1$. A clearer way of phrasing this might be something like "The weight placed on evidence sample n is weighted by $\nu_n = \alpha^{(T-n)}$, where T is the current sample, so that when $\alpha > 1$ this creates a primacy effect and later evidence affects the decision less than the initial evidence, and when $\alpha < 1$ this creates a recency effect, and the observer's decision places greater weight on the more recent evidence than the initial samples."

This comment follows the previous one. The statement has now been changed to:

"Functionally, during accumulation, the current accumulated evidence is weighted by α , before accumulating the next sample, so that when $\alpha > 1$ this creates a primacy effect and later evidence affects the decision less than

the initial evidence, and when $\alpha < 1$ this creates a recency effect, and the observer's decision places greater weight on the more recent evidence than the initial samples",

commented **R3C17a, P18L555**. To clarify this, the description before equation 7 has also been modified (commented at **R3C17b, P18L559** in the revised manuscript).

R3C18 In the Model Fitting section, a formal equation for the log-likelihood that was maximized by the model fitting procedure should be provided.

We did not find an analytic solution for the likelihood function, so it was estimated using Monte Carlo simulation. This has been made clearer in the methods, commented **R3C18, P19L583**:

"As there is no known analytic solution to the likelihood function of the model, the probability of the observer making each response at each stimulus, given the parameters, was numerically estimated using Monte Carlo simulation."

R3C19 p. 19: "The probability of the observer making each response at each stimulus, given the parameters, was numerically estimated by taking 1000 samples from ϵ_n , ..."

This is unclear; please explain this procedure sufficiently clearly and explicitly that someone wishing to reproduce the analysis method could easily and unambiguously do so. There are other parts of the methods that make use of similar language regarding "taking 1000 samples from ϵ_n " that should be similarly clarified.

This has been clarified, commented **R3C19, P19L585**: "was numerically estimated using Monte Carlo simulation."

R3C20 p. 20: "We examined the ideal evidence, ..."

The mathematical expression listed directly following this quote is not the ideal evidence, as it includes the error and temporal bias terms ϵ_n and ν_n . As the authors state in the supplementary material, they define the ideal evidence as evidence uncorrupted by decision noise and temporal bias.

We apologize, this was a typo. It was supposed to read 'absolute'. The change has been commented as **R3C20, P20L620** in the revised manuscript.

R3C21 In the methods, the authors state that the three bounds for defining the four levels of confidence were modeled as exponential functions with the same a and b parameters, but different λ . However, in Fig S2c, the exponentials fit to confidence data for the sample subjects appear to be fits that allow a , b , and λ to vary. It would be helpful to add panels in Fig S2 to show examples of the three fitted λ bounds superimposed on proportional evidence as a function of # of stimuli and confidence level, as this would give a better sense of how the modeling approach relates to the actual data.

We thank the reviewer for this comment, which has also been helpful for addressing comment 08 from reviewer #1. This has been added, commented **R3C21a**, **P05L097** and **R3C21b**, **P05L107**:

Figure S3. Decision bounds and confidence criteria. **a)** The y-axis shows the proportional evidence (disrupted by samples of noise and leak – in this case, $\sigma = 1.5$, $\alpha = 0.9$) which is the accumulated evidence divided by the number of samples the evidence was accumulated over. The dashed lines show the average proportional evidence where the optimal observer chose to respond whilst the solid lines show the fit of an exponentially decreasing function. **b)** The same simulated data as in **a)** is shown with the fit of a power function. The simulations show a steeper decrease than can be captured by the power function (the simulated optimal observer is more cautious with fewer samples and more liberal with more samples). **c)** The y-axis shows the proportional ideal accumulated evidence (not disrupted by noise and leak). The dashed lines show the proportional evidence by the number of samples for each confidence rating (three observers in separate plots). The solid lines show the fit of the exponential function. **d)** Model fitted bounds on the observer’s estimated proportional evidence based on parameters fitted to the confidence decisions. Dotted lines show the average proportional evidence for each confidence rating.

R3C22 The authors also mention that if the lambda value of a higher confidence bound is smaller than that of a lower confidence bound, this can cause issues for fitting the data. Presumably this is because, given that a and b are fixed, smaller lambda for the higher confidence bound would entail smaller proportional evidence at a given level of # of stimuli, which would entail the paradoxical result that less evidence would be required in order to yield the higher confidence rating than for the lower confidence rating-- is that right? It might be helpful to make this point more explicitly in the text. A simple way to avoid this issue would be to enforce a constraint on the lambda parameters such that $\lambda_1 \leq \lambda_2 \leq \lambda_3$. Was that constraint actually enforced? It is not clear given the way the methods text is currently written.

This is correct: it would be paradoxical for an observer to require more evidence for lower confidence. This has been made explicit in the text (commented as **R3C22a**, **P21L649** in revised manuscript):

“If the λ of a higher confidence bound was smaller than the λ of a lower confidence bound, this resulted in negative likelihoods (as it is paradoxical to require less evidence for higher confidence)...” .

Using the Bayesian Adaptive Search, ‘plausible bounds’ were placed on the parameters to avoid the fitting issues, the details of this are commented as **R3C22b**, **P21L650** in the revised manuscript:

“We therefore implemented ‘plausible’ lower and upper bounds on the parameters, based on initial fits to participants where the model was successfully able to apply the bounds. These plausible bounds are used by the Bayesian Adaptive Direct Search to design the initial mesh of the parameter search, and by specifying increasing but overlapping plausible bounds on the λ ’s, the model was able to successfully recover sensible parameters for

all participants, whilst not limiting the model's ability to describe the behaviour of some of the more extreme participants."

Minor points

R3C23 p. 1: "The Type-II decision estimates the likelihood of the Type-I decision, given the accumulated evidence."

The term "likelihood" is ambiguous here. Presumably the authors mean to refer to the likelihood that the Type-I decision is correct.

We thank the reviewer for pointing this out. This has now been fixed, commented as **R3C23 P01L32**:

"The Type-II decision estimates the likelihood that the Type-I decision is correct, given the accumulated evidence."

R3C24 eq 2: in the $p(\theta_1, \dots, \theta_T | \psi)$ term, the subscript on the last theta should be T rather than Tn.

We thank the reviewer for pointing this out. This has now been fixed, commented as **R3C24, P17L538** in revised manuscript.

R3C25 eq 9: "Lamba_n+" should be "Lambda_n"

This is the positive bound, the negative bound is multiplied by -1, this detail is added and commented as **R3C25, P18L569**:

"

$$\Lambda_{n+} = n \times \left(a + b e^{-\frac{n}{\lambda}} \right) \quad (9)$$

For the positive decision bound (the negative bound, $\Lambda_{n-} = -\Lambda_{n+}$). "

R3C26 p. 18: "The likelihood $f(n)$ of responding at sample n was estimated by computing the frequencies, over 1000 particles, ..."

Unclear what "particles" means here.

We thank the reviewer for pointing this out. This has now been fixed, commented as **R3C26, P18L570** in revised manuscript: “1000 samples from ε_n (Monte Carlo simulation)”

R3C27 p. 18: "the mean, T , and variance, $(\sigma_T)^2$, of the non-decision time..."

T should not be used as the mean of the non-decision time since it is previously used to mean the current sample in a trial, and/or the total eventual number of samples in a trial. However, it looks as though this usage of T is replaced with U later on. If that is right, then the T s that appear in the quoted sentence should be replaced with U s.

We thank the reviewer for pointing this out. This has now been fixed, commented as **R3C27, P19L573** in revised manuscript.

R3C28 p. 18: " $g(n; U, \sigma^2)$ is the Gaussian kernel..."

σ^2 is missing the U subscript.

We thank the reviewer for pointing this out. This has now been fixed, commented as **R3C28, P19L577** in revised manuscript

Reviewers' Comments:

Reviewer #1:

Remarks to the Author:

The authors have addressed my main concerns. Congratulations to the authors.

Reviewer #3:

Remarks to the Author:

The authors have very clearly and thoroughly responded to the initial review in a way that largely alleviates the concerns raised and improves the manuscript. I thank the reviewers heartily for their very clear and comprehensive annotation system, which took a lot of work on their part and made my work in evaluating their response much easier. I have only a few remaining points to be addressed, which I will refer to using the authors' system for annotating reviewer comments.

R3C01: I appreciate the authors' in-depth reply here. Their point is well-taken that the current study demonstrates online application of type 2 processes during the ongoing process of type 1 evidence accumulation, in contrast to the majority of extant studies which examine confidence as a post-decisional process. Their results are indeed novel, informative, rigorous, and of great interest and value to the field.

I did not mean to imply that the results are trivial, just that it is possible that some of the results may be strongly influenced by the particular task instructions subjects were given. This point can be expressed something as follows: prior to reading this paper, if someone told me that subjects were instructed to accumulate evidence so as to reach performance targets of 70% vs 85% vs 90% in different experimental conditions, and that they were largely successful in doing so, it would seem to follow very naturally as a prediction that the subjects who are better able to perform this task (i.e. better able to assess their task performance, as reflected in hitting an X% accuracy target) would also be expected to exhibit better post-decision metacognitive sensitivity (i.e. be better able to assess their task performance, as reflected in the type 2 ROC). This is because both tasks fundamentally involve the task of assessing likelihood of being correct at the single-trial level (although again, it is indeed salient to point out that one requires online assessment and the other only requires post-decisional assessment). The only alternative to this would be something like the authors' observation that "If bounds were set based on an estimate of accuracy from some separate learning mechanism, we may not have found a significant relationship between Type-II efficiency and bound efficiency"; however, a finding like that would indeed be very surprising and unexpected.

I think the authors' addition on P13L414, "However, it remains to be tested whether observers utilise their Type-II evidence for bound-setting in other contexts" does not adequately express the nature of the concern. A better phrasing to express the concern would be something like the following: "However, it remains to be tested whether observers utilise their Type-II evidence for bound-setting in other contexts. In particular, in the Stopping task subjects were explicitly instructed to perform the task so as to achieve various performance benchmarks, and doing so presumably requires online monitoring of estimated task performance. Thus, compliance with task instruction may have required subjects to set decision bounds dependent on online Type-II evaluations of task performance. Future research should investigate the role of Type-II processes in Type-I evidence accumulation in cases where a tight relationship between the two is not required or suggested by task instruction." Note, I'm not necessarily saying the authors should insert this text word-for-word, but I do think a slightly more extended discussion of the issue along these lines is called for.

The authors also state "Observers can over- or under-estimate their accuracy but still have good bound efficiency so long as they optimally adapt their bound. This measure focusses on the implementation of the bound, minimising the importance of actually meeting the performance targets." I am not sure I understand the relevance of these points. Bound efficiency is calculated relative to the change in bound a subject should implement in order to reach a given target performance level. Thus, a subject's actual change in bound should ideally be computed with reference to an accurate estimate of their performance level, as it compares to the target performance level. I agree that it is computationally possible for a subject to e.g. systematically underestimate their accuracy but still have excellent bound efficiency-- for instance, perhaps the systematic underestimation of task performance is accompanied by a systematic overestimation of how much the bound needs to be shifted in light of that estimated task performance, such that the two cancel out and the subject lucks into having perfect bound efficiency. But such a systematic underestimation of accuracy would be akin to a conservative type 2 criterion, and bias in type 2 criterion setting is independent from type 2 sensitivity; thus, in this case, we would still expect the subject's excellent bound efficiency to be accompanied by excellent type 2 efficiency. Conversely, if a subject randomly overestimates and underestimates accuracy across trials, this should necessarily translate into both poor bound efficiency and poor type 2 efficiency.

R3C02: To clarify, when I wrote "the computational modeling does not incorporate a mechanism that reflects a possible direct interaction between type-II efficiency and the setting of type-I decision bounds," I wasn't asking about a possible neural mechanism, but rather I meant that there was nothing in the computational model itself corresponding to something like the red arrow in Figure 5. The comment was therefore a general observation about the modeling, which reviewer 1's R1C05 also recognized and provided a specific suggestion for addressing using the existing modeling framework. The analysis conducted for R1C05 directly addresses the issue raised in R3C02 point 1 in a satisfactory way.

R3C06: The authors' response was helpful, but I think there can still be some further clarification of these points in the manuscript. The original puzzle was why the Wilcoxon signed rank tests for the (85% vs 90% Stopping tasks) and (Free vs More tasks) yielded such different results, in spite of the confidence intervals for $p(\text{correct})$ being very similar for these two comparisons (0.80 ± 0.021 vs 0.82 ± 0.023 for the two Stopping conditions, 0.80 ± 0.025 vs 0.82 ± 0.025 for Free vs More). After considering the authors' helpful reply, it now seems to me the reason for this apparent discrepancy comes down to two main points:

1. The statistical tests are conducted on d' , not $p(\text{correct})$, and indeed the magnitude of the mean difference in d' for 85% vs 90% Stopping (1.74 vs 1.92) is roughly double the magnitude of the d' difference for Free (all trials) vs More (1.75 vs 1.85), in spite of the mean differences in $p(\text{correct})$ for these comparisons being roughly equal. This seems to be the primary explanation for the apparent discrepancy.

2. Performance in the More condition ($p(c)=0.82$, $d'=1.85$) was not compared to performance across all 300 Free trials ($p(c)=0.80$, $d'=1.75$), but rather was compared to performance for the subset of Free trials used to create the More trials ($p(c)=?$, $d'=?$).

(The authors also correctly note that the statistical tests were within-subject, and thus the relevant

confidence intervals would be those constructed from the within-subject difference in task performance across conditions, rather than the confidence intervals computed separately for each condition. However, the width of the within-subject confidence intervals they reported in their reply were roughly similar for the 85% vs 90% Stopping comparison and the Free vs More comparison, and so do not seem to be a major factor in explaining why the signed rank tests yield such different results for these comparisons.)

Points (1) and (2) relate to one of the points raised in my initial review, which is that the authors conduct their signed rank tests on d' but summarize task performance with $p(\text{correct})$ rather than d' , which is potentially cause for unnecessary confusion. I understand the rationale for reporting $p(\text{correct})$, but the authors should also report the relevant d' values for all statistical tests comparing d' values. In their revision, they have now added reports of d' mean and CI for the 70%, 85% and 90% Stopping conditions, which helps to shed light on the signed rank tests comparing those conditions. However, they still do not report d' mean and CI for the tests comparing Less vs Free and More vs Free (P5L135-138). d' mean and CI should be added to the reports of these tests.

There is an additional subtlety, which is that the Less vs Free and More vs Free tests (P5L135-138) report $p(\text{correct})$ for the Less and More conditions, but they do not report task performance for the relevant Free conditions in these tests, which is not performance across all 300 Free trials, but rather corresponds to task performance in the relevant 100-trial subsets of the Free condition. This is why, in point (2) above, I marked $p(c)$ and d' for the 'More' subset of the 'Free' trials as unknown-- they are not reported in the manuscript or in the authors' reply. (Similarly, the d' for the More condition was only listed in the reply, and not in the manuscript.)

I don't think it's necessary to report $p(\text{correct})$ for the 'Less' and 'More' subsets of the Free trials. However, the relevant d' means and CIs should be reported for the Less vs Free and More vs Free tests, and this should include reporting the d' data for the 'Less' and 'More' subsets of the Free trials in the relevant statistical tests, since those are the relevant statistical data entering into those tests.

While I now understand this subtlety of the analysis approach, I think its description in the manuscript could still be further clarified, as it remains less clear than the authors' explanation in the reply to reviewers. I recommend rephrasing something along the lines of the following, for maximal clarity:

P4L127: change "certain" to "fixed"

P4L128 - L130: rewrite sentence as "The number of samples presented on each trial was determined relative to how many samples the observer chose to observe for the three repetitions of each pre-defined trial in the 'Free' task."

P4L133: Before the sentence starting with "Performance in the 'Same' condition..." add "We compared performance in the 100 trials of the 'Less,' 'Same,' and 'More' conditions to the corresponding sets of 100 trials in the 'Free' task exhibiting the minimum, median, and maximum number of observed samples across the three pre-defined trial repetitions."

R3C08: The addition on P07L207, " $p(\text{slope} > 0) = 0.026$, with outlier participants removed)" uses a misleading notation, since the p -value for this test does not denote $p(\text{slope} > 0)$ but rather denotes $p(\text{sampled slope at least as extreme as observed value} \mid \text{true slope} = 0)$. I recommend rephrasing as " p -value for slope = 0.026" or something similar.

R3C12: The analysis conducted here is now much clearer. However, it appears that quantitative details of the model comparison are not reported. There is no figure in the main manuscript or supplementary materials showing the quantitative results of the model comparison analysis (e.g. something analogous to Fig 4b), nor is there any report in the text of the quantitative outcome of the model comparison analysis. We are only told which model "won" the model comparison. The manuscript is already very rich with figures and technical detail, so I can understand the impulse towards brevity here. At a minimum, though, we should be given some indication in the text regarding the quantitative outcome of the model comparison; perhaps a condensed way to report this would be to report some quantitative comparison between the model fit for the best-fitting and second-best-fitting model, supplemented with a conceptual interpretation of that quantitative comparison that gives the reader a sense of how much more strongly the best-fitting model was supported than the second-best. Alternative ways of reporting the quantitative model comparison results are fine too, as long as the reader is given some sense of the actual quantitative strength of support for the best-fitting model relative to the others.

Also, on P20L623: "an example is shown in Figure S2." This should refer to Figure S3 due to the addition of the new Figure S1. It appears there are other errors of this nature as well, e.g. P18L568 should refer to Figure S4, not Figure S3. The authors should double-check all references to supplementary figures to ensure they are correct.

R3C21: Why do the dashed lines in panels c and d differ? I surmise it is something to do with (c) plotting "proportional ideal evidence" and (d) plotting "proportional evidence," but it's still a little unclear. Can this be clarified, and brief justification / explanation be given for why (c) and (d) plot different quantities?

signed,
Brian Maniscalco

We thank the reviewers for their continued support in improving this manuscript. Below, we have outlined how we have addressed the remaining concerns of Reviewer #3, in relation to their previous comments (numbered 1, 2, 6, 8, 12, and 21). We hope that this is satisfactory and thank the reviewer for their careful attention in reviewing this manuscript. In our response, we have included the initial comments and our initial response, for clarity, these are in green text colour. The previous comments are labelled as before, the new comments are labelled beginning **R3B**, such that **R3C01** refers to Reviewer #3's comment from the first review and **R3BC01** refers to Reviewer #3's comment relating to **R3C01** from the second review. The new comments are in blue text colour, with our response in black. Critical suggestions from the reviewer's new comments are highlighted in bold.

Reviewer #3 (Remarks to the Author):

The authors have very clearly and thoroughly responded to the initial review in a way that largely alleviates the concerns raised and improves the manuscript. I thank the reviewers heartily for their very clear and comprehensive annotation system, which took a lot of work on their part and made my work in evaluating their response much easier. I have only a few remaining points to be addressed, which I will refer to using the authors' system for annotating reviewer comments.

R3C01 To my mind, the most interesting aspect of this manuscript is the demonstration of a link between optimality in metacognition and optimality in perceptual decision-making, which could be interpreted as potentially being a tell-tale sign of a type-II process exerting top-down control on a type-I process. However, one consideration that significantly moderates my enthusiasm is that the 'Stopping' task requires subjects to provide a type-I response once they've reached a certain threshold of estimated probability that their response will be correct, given the evidence accumulated thus far. But this means that the very nature of the task requires subjects to yoke the type-I response ("I think the stimuli were drawn from the blue distribution") to a type-II judgment ("I will only provide my type-I response once I think I'm 80% likely to be correct"). So, perhaps it is not so surprising that the subjects who are poor at distinguishing correct from incorrect responses with a 4-point confidence rating scale (poor type-II efficiency) are also poor at producing an absolute probability estimate of how likely they are to be correct (poor bound efficiency). More broadly, I worry that the type-I / type-II correlation could possibly be a contrived consequence of the task design and instructions in the 'Stopping' task, rather than reflecting a more general relationship that can be observed in more naturalistic or open-ended task settings that do not so explicitly encourage subjects to yoke type-I responses to type-II judgments in this way. I do not think this is a fatal objection or a completely trivializing observation, but it should be addressed somewhere in the Discussion.

Initial response: We thank the reviewer for outlining this concern. Indeed, it has been difficult to sufficiently emphasize that our results are by no means trivial, although some readers may initially find the relationship unsurprising. In our manuscript we provide the first test of the relationship between the observers' ability to efficiently accumulate perceptual evidence (bound efficiency) and their metacognitive insight into their own perceptual performance (Type-II efficiency). We show this by explicitly setting target performance levels and measuring how well observers adjust their bounds in the face of these performance targets. If there were no relationship between Type-II and bound efficiency in this task, then one would not expect to find the relationship in tasks where the need to estimate accuracy for the purpose of implementing bounds is less explicit. We interpret that the reviewer's main concern is that these results may not generalise to more naturalistic tasks where the observer is not explicitly asked to set their bounds according to a performance target. It is true that we cannot demonstrate this with our data, and we have added a comment to the discussion to emphasise that this should be examined in future research (**R3C01a**, **P13L414**, see quoted text below).

There are two ways a reader could interpret the relationship between Type-II and bound efficiency as trivial. First, it could be that some observers are better across all experimental measures (because of motivational factors, or their ability to concentrate throughout). We address this first issue with two comparisons: (i) the relationship between observer's Type-I sensitivity in the 'Replay' task and bound efficiency in the 'Stopping' task, and (ii) the relationship between Type-II efficiency in the 'Replay' task and the magnitude of the suboptimalities in evidence accumulation in the 'Stopping' task. There was no relationship between these variables, so it is not the case that there is a general correlation between different performance measures across tasks. This we emphasise in the discussion, commented **R3C01b**, **P13L397**, quoted below.

Second, it could be that observers are only using their Type-II evidence to set and maintain their bounds efficiently because we instructed them to (and otherwise there would be no relationship). This second issue is conceptual. At the surface, one could suggest that it is unsurprising that observers who are poor at discriminating correct from incorrect responses with confidence ratings should also be poor at estimating the likelihood that they are correct, and therefore show poor bound efficiency. But the broader theoretical context suggests the relationship is not so simple. The current normative description of confidence suggests (at least implicitly) that it is not computed online: that only after the decision is made does confidence take the perceptual evidence and convert it into an estimate of the likelihood of being correct. In order to use confidence evidence to implement bounds, confidence must be computed online. If confidence were computed only after the decision, we may not have found a significant relationship between Type-II efficiency and bound efficiency. The current theories of how observers implement bounds suggests the involvement of learning mechanisms to balance speed and accuracy. The focus of the discussion has been on how observers are able to flexibly adjust their bounds in the face of different costs or speed pressures, whilst the importance of estimating accuracy has been minimised. If bounds were set based on an estimate of accuracy from some separate learning mechanism, we may not have found a significant relationship between Type-II efficiency and bound efficiency. Further, our analysis describes observers' ability to efficiently adjust their bounds: bound efficiency is calculated as the change in bound between 70% and 85% target performance conditions, divided by the ideal change in the bound. This measure emphasizes

perceptual precision over perceptual accuracy. Observers can over- or under-estimate their accuracy but still have good bound efficiency so long as they optimally adapt their bound. This measure focusses on the implementation of the bound, minimising the importance of actually meeting the performance targets. We have explained this more fully in the results, commented **R3C01c, P06L196**:

“‘Bound efficiency’ was then calculated as the change in bound across conditions that the observer actually implemented, divided by the change in bound they should have implemented if they were to reach the target performance levels. This expected behaviour was obtained by simulations given the other suboptimalities affecting observers’ performance (see **Methods** for more details). In this way, an observer with high bound-efficiency could over- or under-estimate their performance (poor accuracy relative to the target performance levels), though they still *adjust* their bounds appropriately in the face of different performance targets (good precision).”

Thus, the strong relationship between bound efficiency and Type-II efficiency suggests that the ability to flexibly adjust bounds strongly relies on the representation of the likelihood of being correct that is generated by the same metacognitive system as post-decisional confidence judgements. We measure Type-II efficiency and bound efficiency in separate tasks, in separate sessions. We show that Type-II decisions are based on a partially dissociable accumulation process to Type-I decisions, with dissociable effects on pupil size. Yet these distinct measures, based on distinct processing, measured in separate sessions, are strongly correlated.

Here is the relevant section of the discussion:

“Despite all these differences between Type-I and Type-II evidence accumulation, there was a surprisingly strong correlation between observers’ Type-II efficiency and their ability to efficiently set and maintain Type-I decision bounds. It is unlikely that this correlation emerges indirectly due to some common underlying variable, such as the observers’ motivation in the experiment in general, because of the lack of a relationship between observer’s Type-I sensitivity (which should also vary with task commitment) in the ‘Replay’ task and bound efficiency in the ‘Stopping’ task. Likewise, there was no relationship between Type-II efficiency in the ‘Replay’ task and the magnitude of the suboptimalities in evidence accumulation in the ‘Stopping’ task. The strongest interpretation of the correlation is that there is a causal relationship between Type-II efficiency and bound efficiency: Type-II evidence is being used to set and maintain boundaries on Type-I evidence accumulation (though a test for causality is left for future research). This ‘metacognitive control’ has been previously suggested to operate in a post-decisional manner: the current confidence will influence future Type-I evidence accumulation^{37,38,39}. Our interpretation goes further by postulating that the Type-II process is acting on the Type-I evidence accumulation online, as accumulation occurs. Indeed, we were able to reproduce observers’ bound efficiency by simulating bounds enacted based on the Type-II evidence estimate of the probability of correct responses (**Supplementary Materials, Figure S5**). This kind of interaction would readily explain how observers actively seek more information when they are uncertain⁴⁰, and how observers can integrate loss functions into their Type-I decisions⁴¹. In addition, this kind of relationship offers a mechanism by which decision bounds could be implemented at the neural level, which is supported by evidence showing activity in the dlPFC is modulated

according to changes in the speed-accuracy trade-off^{42,43,44} and according to metacognitive confidence^{45,46}. However, it remains to be tested whether observers utilise their Type-II evidence for bound-setting in other contexts.”

We hope that the reviewer will find these additions useful.

R3BC01: I appreciate the authors' in-depth reply here. Their point is well-taken that the current study demonstrates online application of type 2 processes during the ongoing process of type 1 evidence accumulation, in contrast to the majority of extant studies which examine confidence as a post-decisional process. Their results are indeed novel, informative, rigorous, and of great interest and value to the field.

I did not mean to imply that the results are trivial, just that it is possible that some of the results may be strongly influenced by the particular task instructions subjects were given. This point can be expressed something as follows: prior to reading this paper, if someone told me that subjects were instructed to accumulate evidence so as to reach performance targets of 70% vs 85% vs 90% in different experimental conditions, and that they were largely successful in doing so, it would seem to follow very naturally as a prediction that the subjects who are better able to perform this task (i.e. better able to assess their task performance, as reflected in hitting an X% accuracy target) would also be expected to exhibit better post-decision metacognitive sensitivity (i.e. be better able to assess their task performance, as reflected in the type 2 ROC). This is because both tasks fundamentally involve the task of assessing likelihood of being correct at the single-trial level (although again, it is indeed salient to point out that one requires online assessment and the other only requires post-decisional assessment). The only alternative to this would be something like the authors' observation that "If bounds were set based on an estimate of accuracy from some separate learning mechanism, we may not have found a significant relationship between Type-II efficiency and bound efficiency"; however, a finding like that would indeed be very surprising and unexpected.

I think the authors' addition on P13L414, "However, it remains to be tested whether observers utilise their Type-II evidence for bound-setting in other contexts" does not adequately express the nature of the concern. **A better phrasing to express the concern would be something like the following: "However, it remains to be tested whether observers utilise their Type-II evidence for bound-setting in other contexts. In particular, in the Stopping task subjects were explicitly instructed to perform the task so as to achieve various performance benchmarks, and doing so presumably requires online monitoring of estimated task performance. Thus, compliance with task instruction may have required subjects to set decision bounds dependent on online Type-II evaluations of task performance. Future research should investigate the role of Type-II**

processes in Type-I evidence accumulation in cases where a tight relationship between the two is not required or suggested by task instruction." Note, I'm not necessarily saying the authors should insert this text word-for-word, but I do think a slightly more extended discussion of the issue along these lines is called for.

The authors also state "Observers can over- or under-estimate their accuracy but still have good bound efficiency so long as they optimally adapt their bound. This measure focusses on the implementation of the bound, minimising the importance of actually meeting the performance targets." I am not sure I understand the relevance of these points. Bound efficiency is calculated relative to the change in bound a subject should implement in order to reach a given target performance level. Thus, a subject's actual change in bound should ideally be computed with reference to an accurate estimate of their performance level, as it compares to the target performance level. I agree that it is computationally possible for a subject to e.g. systematically underestimate their accuracy but still have excellent bound efficiency-- for instance, perhaps the systematic underestimation of task performance is accompanied by a systematic overestimation of how much the bound needs to be shifted in light of that estimated task performance, such that the two cancel out and the subject lucks into having perfect bound efficiency. But such a systematic underestimation of accuracy would be akin to a conservative type 2 criterion, and bias in type 2 criterion setting is independent from type 2 sensitivity; thus, in this case, we would still expect the subject's excellent bound efficiency to be accompanied by excellent type 2 efficiency. Conversely, if a subject randomly overestimates and underestimates accuracy across trials, this should necessarily translate into both poor bound efficiency and poor type 2 efficiency.

We thank the reviewer for this in-depth consideration of the implications of our experimental results. We have paraphrased the suggested insertion, keeping in mind the word-limit of *Nature Communications*. We have understood that the reviewer's main concern is that it should be specifically noted that task instructions could have influenced the way observers do the task. We therefore now state (**P13L419**, commented **R3BC01**):

"However, it remains to be tested whether observers utilise their Type-II evidence for bound-setting in other contexts, in particular, in contexts where the tight relationship between Type-II evidence and bound setting is not suggested in the task instructions."

R3C02 Although I do not count this against the current manuscript, some gaps in the logic of the overall argument that could be filled out in future research are: (1) the computational modeling does not incorporate a mechanism that reflects a possible direct interaction between type-II

efficiency and the setting of type-I decision bounds; (2) ideally, there would be a way to test for a causal relationship between type-II processing and type-I decision making. It may be worth discussing these points briefly in the Discussion as areas for future work to address.

Initial response: These are indeed essential points for future research, which are all now present in the discussion of the revised manuscript. Point 1 is discussed at **P13L411**, commented **R3C02a**. The simulations suggested by reviewer #1 (**R1C05**) suggests that the bound could be set directly on the estimates based on the Type-II evidence, though this does not directly clarify the neural mechanisms. Point 2 has now been noted in the discussion, commented **R3C02b**, **P13L404**, incorporated into the quote above.

R3BC02: To clarify, when I wrote "the computational modeling does not incorporate a mechanism that reflects a possible direct interaction between type-II efficiency and the setting of type-I decision bounds," I wasn't asking about a possible neural mechanism, but rather I meant that there was nothing in the computational model itself corresponding to something like the red arrow in Figure 5. The comment was therefore a general observation about the modeling, which reviewer 1's R1C05 also recognized and provided a specific suggestion for addressing using the existing modeling framework. **The analysis conducted for R1C05 directly addresses the issue raised in R3C02 point 1 in a satisfactory way.**

We thank the reviewer for this clarification, indeed the combined input of Reviewer #1 and Reviewer #3 has been invaluable in strengthening the manuscript in this regard.

R3C06 Perhaps related to the above, I found it odd that the reported confidence intervals for proportion correct in the 85% and 90% conditions of the Stopping task (0.80 ± 0.021 and 0.82 ± 0.023) were nearly identical to those of the 'Free' and 'More' tasks (0.80 ± 0.025 and 0.82 ± 0.025), and yet the results of the Wilcoxon signed rank tests yielded a significant difference in the former case ($Z=2.35$, $p=0.037$) but was nowhere close to significant in the latter ($Z=0.04$, $p=0.68$). I realize the statistical tests were performed on d' rather than proportion correct, and that the 'Free' and 'More' data were slightly noisier, but the stark difference in the Wilcoxon tests is still somewhat surprising. Do these two statistical comparisons look more similar if they are assessed with a parametric test rather than non-parametric? This is not an entirely trivial point since the authors interpret the lack of statistically significant difference in task performance between the 'Free' and 'More' conditions as one piece of evidence for the claim that subjects set covert decision bounds that terminate evidence accumulation prior to perceiving all the additional stimuli in the 'More' condition. Even if an alternative test did reveal a significant difference between the 'Free' and 'More' conditions, I don't think it would make all that much of a

difference to the authors' overall argument, but it would nonetheless be informative to get clearer on all this.

Initial response: This is a meaningful concern, also raised by reviewer #1 (**R1C03**), which is due to a lack of sufficient description in the original manuscript. Indeed, we left an important detail to the methods section (**P15L462**, and **P15L488**), which can explain the apparent discrepancy pointed out by the reviewers. We have tried to explain this detail better in the results now, whilst maintaining the brevity required by *Nature Communications* (commented **R3C06a**, **P03L81**; **R3C06b**, **P04L129**; **R3C06c**, **P04L134**). This detail is that the same 100 pre-defined trials for each participant were repeated throughout the entire experiment. In the 'Free' task, observers completed each trial three times. Therefore, for each trial, there were three responses (3 repeats) and three 'number of samples' ('Less', 'Same', and 'More' conditions). For the 'Less' condition of the 'Replay' task, we took the *minimum* number of samples minus 2 samples. For the 'Same' condition, we took the *median* number of samples. And for the 'More' condition, we took the *maximum* number of samples plus 4 samples. For clarity, an explicit example of this procedure has been added to the methods, commented **R3C06d** on **P15L490**:

"For example, if the observer chose to respond after 4, 5, and 10 samples on the three repeats of one pre-defined trial, they would be presented with 2, 5, and 14 samples in the 'Less', 'Same', and 'More' conditions of the 'Replay' task respectively."

The comparison of d' was not conducted across all trials of the 'Free' task (300), but for the 100 trials chosen for each condition ('Less', 'Same', and 'More') in the 'Replay' task. Thus, the comparison asks if precisely 4 additional samples improve d' , not if sensitivity is improved in the 'More' condition relative to the 'Free' task taken as a whole. This has been clarified and commented as **R3C06e**, **P05L135**:

"In the 'Less' condition, two fewer samples corresponded to a substantial decrease in performance within subjects ($M = 0.69 [\pm 0.025]$, Z ('Less' vs. 'Free' d') = 3.51, $p_{\text{bonf}^*3} < 0.001$) but in the 'More' condition, four additional samples did not significantly improve performance within subjects ($M = 0.82 [\pm 0.025]$; $Z = 0.04$, $p = 0.68$)."

Importantly, when we compare d' in the 'More' condition to d' calculated across all trials of the 'Free' task, the difference is also not significant ($Z = 1.34$, $p = 0.18$). Note, also, that because our test is a within-subject comparison, it is important to consider the standard error of the difference. In the 'Stopping' task, d' was on average 1.74 for an 85% target and 1.92 for a 90% target, corresponding to an average difference of 0.18 and a 95% within-subject confidence of 0.064 (yielding a significant difference). In the 'More' condition, d' was on average 1.85 and in the 'Free' task (all trials) it was on average 1.75, corresponding to an average difference of 0.095 (about half the above) and a 95% within-subject confidence of 0.068. In other words, there is a substantially smaller within-subject effect size for the difference between the 'More' condition and the 'Free' task than between the 90% and the 85% target conditions within the 'Free' task.

Using a parametric t-test, neither comparison is significant (all trials: $t(19) = 1.42$, $p = 0.17$; matched +4 trials: $t(19) = 0.78$, $p = 0.44$). We hope this alleviates the reviewer's concerns.

R3BC06: The authors' response was helpful, but I think there can still be some further clarification of these points in the manuscript. The original puzzle was why the Wilcoxon signed rank tests for the (85% vs 90% Stopping tasks) and (Free vs More tasks) yielded such different results, in spite of the confidence intervals for $p(\text{correct})$ being very similar for these two comparisons (0.80 ± 0.021 vs 0.82 ± 0.023 for the two Stopping conditions, 0.80 ± 0.025 vs 0.82 ± 0.025 for Free vs More). After considering the authors' helpful reply, it now seems to me the reason for this apparent discrepancy comes down to two main points:

1. The statistical tests are conducted on d' , not $p(\text{correct})$, and indeed the magnitude of the mean difference in d' for 85% vs 90% Stopping (1.74 vs 1.92) is roughly double the magnitude of the d' difference for Free (all trials) vs More (1.75 vs 1.85), in spite of the mean differences in $p(\text{correct})$ for these comparisons being roughly equal. This seems to be the primary explanation for the apparent discrepancy.

2. Performance in the More condition ($p(c)=0.82$, $d'=1.85$) was not compared to performance across all 300 Free trials ($p(c)=0.80$, $d'=1.75$), but rather was compared to performance for the subset of Free trials used to create the More trials ($p(c)=?$, $d'=?$).

(The authors also correctly note that the statistical tests were within-subject, and thus the relevant confidence intervals would be those constructed from the within-subject difference in task performance across conditions, rather than the confidence intervals computed separately for each condition. However, the width of the within-subject confidence intervals they reported in their reply were roughly similar for the 85% vs 90% Stopping comparison and the Free vs More comparison, and so do not seem to be a major factor in explaining why the signed rank tests yield such different results for these comparisons.)

Points (1) and (2) relate to one of the points raised in my initial review, which is that the authors conduct their signed rank tests on d' but summarize task performance with $p(\text{correct})$ rather than d' , which is potentially cause for unnecessary confusion. I understand the rationale for reporting $p(\text{correct})$, but the authors should also report the relevant d' values for all statistical tests comparing d' values. In their revision, they have now added reports of d' mean and CI for the 70%, 85% and 90% Stopping conditions, which helps to shed light on the signed rank tests comparing those conditions. However, they still do not report d' mean and CI for the tests comparing Less vs Free and More vs Free (P5L135-138). **d' mean and CI should be added to the reports of these tests.**

There is an additional subtlety, which is that the Less vs Free and More vs Free tests (P5L135-138) report $p(\text{correct})$ for the Less and More conditions, but they do not report task performance for the relevant Free conditions in these tests, which is not performance across all 300 Free trials, but rather corresponds to task performance in the relevant 100-trial subsets of the Free condition. This is why, in point (2) above, I marked $p(c)$ and d' for the 'More' subset of the 'Free' trials as unknown-- they are not reported in the manuscript or in the authors' reply. (Similarly, the d' for the More condition was only listed in the reply, and not in the manuscript.)

I don't think it's necessary to report $p(\text{correct})$ for the 'Less' and 'More' subsets of the Free trials. However, the relevant d' means and CIs should be reported for the Less vs Free and More vs Free tests, and this should include reporting the d' data for the 'Less' and 'More' subsets of the Free trials in the relevant statistical tests, since those are the relevant statistical data entering into those tests.

While I now understand this subtlety of the analysis approach, I think its description in the manuscript could still be further clarified, as it remains less clear than the authors' explanation in the reply to reviewers. I recommend rephrasing something along the lines of the following, for maximal clarity:

P4L127: change "certain" to "fixed"

P4L128 - L130: rewrite sentence as "The number of samples presented on each trial was determined relative to how many samples the observer chose to observe for the three repetitions of each pre-defined trial in the 'Free' task."

P4L133: Before the sentence starting with "Performance in the 'Same' condition..." add "We compared performance in the 100 trials of the 'Less,' 'Same,' and 'More' conditions to the corresponding sets of 100 trials in the 'Free' task exhibiting the minimum, median, and maximum number of observed samples across the three pre-defined trial repetitions."

We thank the reviewer for pointing out these missing details, and for their help in clarifying the text. The requested summary statistics are now written alongside the statistical comparison (**P5L138**, commented **R3BC06A**):

"In the 'Less' condition, two fewer samples corresponded to a substantial decrease in performance within subjects ('Less' $d' = 1.01$; 'Free' d' on 'Less' trials = 1.50; Mean within subjects difference = 0.49 [± 0.13], $Z = 3.51$, $p_{\text{bonf}^*3} < 0.001$) but in the 'More' condition, four additional samples did not significantly improve performance

within subjects ('More' $d' = 1.85$ 'Free' d' on 'More' trials = 1.77; Mean within subjects difference = -0.08 [± 0.13]; $Z = 0.04$, $p = 0.68$)."

We have implemented the specific changes requested to the text, commented **R3BC06B**, **P4L127**; **R3BC06C**, **P4L128 - L130**; though we kept "observer chose to respond to" over "observer chose to observe"); **R3BC06D**, **P4L133**.

The full section now reads:

"The same trials were then repeated to observers in the 'Replay' task, completed immediately after the 'Free' task, but in the 'Replay' task observers were presented with a fixed number of samples and could only respond after the response cue. The number of samples presented on each trial was determined relative to how many samples the observer chose to respond to for the three repetitions of each pre-defined trial in the 'Free' task. There were three intermixed conditions: 'Less' (-2 samples from the minimum), 'Same' (same number of samples as the median), and 'More' (+4 samples from the maximum). This resulted in a median number of samples of 5.4, 10.4, and 18.5 in the 'Less', 'Same', and 'More' conditions respectively. We compared performance in the 100 trials of the 'Less,' 'Same,' and 'More' conditions to the corresponding sets of 100 trials in the 'Free' task exhibiting the minimum, median, and maximum number of observed samples across the three pre-defined trial repetitions."

R3C08 The result plotted in Fig 3d as described in the figure legend is different from that reported in the main text of the Results, apparently due to omitting two subjects from analysis in the former but not the latter. Why the discrepancy? It is fine to only show the single result in Fig 3d, but both results should be described in the Results, with an expanded justification for omitting two subjects in the modified analysis-- the justification provided in the figure legend is not entirely clear.

Additionally, presumably the p-value reported in the figure legend pertains to the statistical significance of the slope of the fitted equation, is that right? Please clarify in the legend.

Initial response: This is correct: the p-value pertains to the significance of the slope of the fitted equation. The discrepancy is due to whether the outliers are included or not. As requested, the result has been added to the main text (commented as **R3C08a**, **P07L207** in revised manuscript) and the statement in the legend has been clarified (**R3C08b**, **P07L226**):

"Each observer's bound efficiency (x-axis) by their Type-II efficiency (y-axis) and the line minimising the perpendicular distance from these points, here shown excluding two outlier participants (open circles) whose optimal bound was estimated using maximum performance rather than 85% correct ($y = 1.5x + 0.2$, $p(\text{slope}>0) = 0.026$, with outlier participants removed)."

R3BC08: The addition on P07L207, "p(slope>0) = 0.026, with outlier participants removed)" uses a misleading notation, since the p-value for this test does not denote p(slope>0) but rather denotes p(sampled slope at least as extreme as observed value | true slope = 0). **I recommend rephrasing as "p-value for slope = 0.026" or something similar.**

We thank the reviewer for this suggestion, and have implemented this change at **P8L230**, commented **R3BC08**.

R3C12 p. 9: "...we found both σ and α to correlate when allowing model parameters to vary independently across the Type-I and Type-II evidence accumulation processes..."

This is the only mention in the manuscript of a modeling approach that directly reflects the model structure of Fig 5. However, the given explanation of the modeling approach is brief and not entirely clear, and it is not described more fully in the Methods. Presumably what the authors mean is that they modeled two independent evidence accumulation processes, call them z_I and z_{II} , such that type-I response was determined by z_I crossing the covert bound, whereas confidence was determined by where z_{II} fell relative to the three confidence bounds. Is that right? It is also not clear when the readout of confidence from z_{II} occurred in this modeling approach. Did it occur at the same time as z_I crossed the decision bound, or did it occur after the last presentation of the sample? It seems appropriate to give a fuller characterization of this modeling approach in the Methods.

Initial response: We thank the reviewer for highlighting this point that was not completely clear. In the terminology of the reviewer, we fitted models where z_I and z_{II} were the same and different. For models where z_I and z_{II} were different, we either fit an entirely independent set of parameters, or used a 'partially correlated' design, where Type-I parameters were fixed, but the Type-II could incur additional noise and/or a separate leak (and all combinations). Model comparison suggested the most parsimonious explanation of the data was one of these partially correlated models (**Figure 5**). We tested whether there was a bound on Type-II accumulation (**Figure 4B**) and found no evidence for one, so z_{II} incorporated evidence from all samples. These points have now been clarified, commented as **R3C12**, **P21L636**:

"First, a model was fit using the same z for Type-I and Type-II responses. Entirely separate parameters (leading to independent z for Type-I and Type-II responses) were fit in the 'parallel' model, where the Type-I parameters were fixed to those fit to only the Type-I responses. In partially correlated models some parameters for the Type-II z were fixed to be the same as those affecting the Type-I z . These models compared all combinations of fixed/varied noise and leak, and compared whether additional noise was added with each sample of evidence, or

a single sample of noise irrespective of sequence length. Model comparison showed a partially correlated model, where Type-II z is affected by additional noise and a different leak, best accounted for Type-I and Type-II responses. We then compared models in which the observer accumulates Type-II evidence over all samples and models implementing a bound on Type-II evidence accumulation (either the same bound as the Type-I bound, or an independent bound). There was no evidence for a Type-II accumulation bound (**Figure 4b**); Type-II z accumulated evidence across all presented samples.”

R3BC12: The analysis conducted here is now much clearer. However, it appears that quantitative details of the model comparison are not reported. There is no figure in the main manuscript or supplementary materials showing the quantitative results of the model comparison analysis (e.g. something analogous to Fig 4b), nor is there any report in the text of the quantitative outcome of the model comparison analysis. We are only told which model "won" the model comparison. The manuscript is already very rich with figures and technical detail, so I can understand the impulse towards brevity here. **At a minimum, though, we should be given some indication in the text regarding the quantitative outcome of the model comparison;** perhaps a condensed way to report this would be to report some quantitative comparison between the model fit for the best-fitting and second-best-fitting model, supplemented with a conceptual interpretation of that quantitative comparison that gives the reader a sense of how much more strongly the best-fitting model was supported than the second-best. Alternative ways of reporting the quantitative model comparison results are fine too, as long as the reader is given some sense of the actual quantitative strength of support for the best-fitting model relative to the others.

Also, on P20L623: "an example is shown in Figure S2." This should refer to Figure S3 due to the addition of the new Figure S1. It appears there are other errors of this nature as well, e.g. P18L568 should refer to Figure S4, not Figure S3. The authors should double-check all references to supplementary figures to ensure they are correct.

We thank the reviewer for suggesting this additional clarification. The details of the model comparison have been added to the relevant section of the manuscript (**P21L642**, commented **R3BC12**):

“First, a model was fit using the same z for Type-I and Type-II responses. Entirely separate parameters (leading to independent z for Type-I and Type-II responses) were fit in the ‘parallel’ model, where the Type-I parameters were fixed to those fit to only the Type-I responses. In partially correlated models some parameters for the Type-II z were fixed to be the same as those affecting the Type-I z . These models compared all combinations of fixed/variable noise and leak, and compared whether additional noise was added with each sample of evidence, or

a single sample of noise irrespective of sequence length. Model comparison showed a partially correlated model, where Type-II z is affected by additional noise and a different leak, best accounted for Type-I and Type-II responses: the exceedance probability of this model over the model fit using the same z for Type-I and Type-II responses was $x_p > 0.99$; the exceedance probability over the parallel model was $x_p > 0.999$; the exceedance probability over the next best partial model (a model with leak fixed to the Type-I leak) was $x_p = 0.54$. We then compared models in which the observer accumulates Type-II evidence over all samples and models implementing a bound on Type-II evidence accumulation (either the same bound as the Type-I bound, or an independent bound). There was no evidence for a Type-II accumulation bound (**Figure 4b**); Type-II z accumulated evidence across all presented samples.”

We have now been through the manuscript to ensure the correct Supplementary Figures are referred to.

R3C21 In the methods, the authors state that the three bounds for defining the four levels of confidence were modeled as exponential functions with the same a and b parameters, but different λ . However, in Fig S2c, the exponentials fit to confidence data for the sample subjects appear to be fits that allow a , b , and λ to vary. It would be helpful to add panels in Fig S2 to show examples of the three fitted λ bounds superimposed on proportional evidence as a function of # of stimuli and confidence level, as this would give a better sense of how the modeling approach relates to the actual data.

Initial response: We thank the reviewer for this comment, which has also been helpful for addressing comment 08 from reviewer #1. This has been added, commented **R3C21a**, **P05L097** and **R3C21b**, **P05L107**:

Figure S3. Decision bounds and confidence criteria. **a)** The y-axis shows the proportional evidence (disrupted by samples of noise and leak – in this case, $\sigma = 1.5$, $\alpha = 0.9$) which is the accumulated evidence divided by the number of samples the evidence was accumulated over. The dashed lines show the average proportional evidence where the optimal observer chose to respond whilst the solid lines show the fit of an exponentially decreasing function. **b)** The same simulated data as in **a)** is shown with the fit of a power function. The simulations show a steeper decrease than can be captured by the power function (the simulated optimal observer is more cautious with fewer samples and more liberal with more samples). **c)** The y-axis shows the proportional ideal accumulated evidence (not disrupted by noise and leak). The dashed lines show the proportional evidence by the number of samples for each confidence rating (three observers in separate plots). The solid lines show the fit of the exponential function.

d) Model fitted bounds on the observer's estimated proportional evidence based on parameters fitted to the confidence decisions. Dotted lines show the average proportional evidence for each confidence rating.

R3BC21: Why do the dashed lines in panels c and d differ? I surmise it is something to do with (c) plotting "proportional ideal evidence" and (d) plotting "proportional evidence," but it's still a little unclear. **Can this be clarified, and brief justification / explanation be given for why (c) and (d) plot different quantities?**

We thank the reviewer for requesting this additional clarification. The proportional evidence is an estimate of the evidence used by the observer to make their confidence judgements (based on the computational modelling) divided by the number of samples. The evidence used by the observer is the ideal evidence (ℓ_n , Eq 4 in the manuscript), corrupted by noise and leak. Thus, the data for the dotted lines in d) is the data for the dotted lines in c), but corrupted by noise and leak according to Eq 8 in the manuscript, using the best-fitting σ and α values for the confidence decisions of each observer. The figure legend (**P5L104** of the Supplementary materials, commented **R3BC21**) now reads:

“c) The y-axis shows the proportional ideal accumulated evidence ($|\ell_n|$, see equation 4 in manuscript, which describes the decision evidence undisrupted by noise and leak). The dashed lines show the proportional evidence by the number of samples for each confidence rating (three observers in separate plots). The solid lines show the fit of the exponential function. d) Solid lines show the model fitted bounds on the observer's estimated proportional evidence. Dotted lines show the average proportional evidence for each confidence rating. The proportional evidence is the proportional ideal evidence disrupted by noise and leak according to equation 8 of the manuscript. This was estimated using the best-fitting parameters for describing the confidence ratings of the observer.”